# Multispecies characterization of immature neurons in the mammalian amygdala reveals their expansion in primates

Marco Ghibaudi[1,2], Chiara La Rosa[1], Nikita Telitsyn[1,2], Jean-Marie Graïc[3], Chris G. Faulkes[4], Chet C. Sherwood[5], Luca Bonfanti [1,2]*

1 Neuroscience Institute Cavalieri Ottolenghi (NICO), Orbassano, Italy, 2 Department of Veterinary Sciences, University of Turin, Torino, Italy, 3 Department of Comparative Biomedicine and Food Science, University of Padova, Legnaro, Padova, Italy, 4 School of Biological and Behavioural Sciences, Queen Mary University of London, London, United Kingdom, 5 Department of Anthropology and Center for the Advanced Study of Human Paleobiology, The George Washington University, Washington, DC, United States of America

* luca.bonfanti@unito.it

## Abstract

Structural changes involving new neurons can occur through stem cell-driven neurogenesis, and through incorporation of late-maturing "immature" neurons into networks, namely undifferentiated neuronal precursors frozen in a state of arrested maturation. The latter have been found in the cerebral cortex and are particularly abundant in large-brained mammals, covarying with the size of the brain and cortex. Similar cells have been described in the amygdala of some species, although their features and interspecies variation remain poorly understood. Here, their occurrence, number, morphology, molecular expression, age-related changes, and anatomical distribution in amygdala subdivisions were systematically analyzed in eight diverse mammalian species (including mouse, naked mole rat, rabbit, marmoset, cat, sheep, horse, and chimpanzee) widely differing in neuroanatomy, brain size, life span, and socioecology. We identify converging evidence that these amygdala cells are immature neurons and show marked phylogenetic variation, with a significantly greater prevalence in primates. The immature cells are largely located within the amygdala's basolateral complex, a region that has expanded in primate brain evolution in conjunction with cortical projections. In addition, amygdala immature neurons also appear to stabilize in number through adulthood and old age, unlike other forms of plasticity that undergo marked age-related reduction. These results support the emerging view that large brains performing complex socio-cognitive functions rely on wide reservoirs of immature neurons.

## Introduction

In recent years, new interest in brain plasticity has been raised by observations of populations of "immature" or "dormant" neurons (INs) consisting of prenatally

**Data availability statement:** All relevant data are within the paper and its Supporting Information files.

**Funding:** The present work was supported by Progetto Trapezio - Compagnia di San Paolo (grant 67935-2021.2174), Fondazione CRT - Cassa di Risparmio di Torino (grant RF=2022.0618), and PRIN2022 (Ministero dell'Università e della Ricerca; grant 2022LB4X3N) to LB; National Science Foundation (grants EF-2021785, DRL-2219759), and National Institutes of Health (grants NS092988, AG067419) to CCS. None of the funders played a role in the study design, data collection and analysis, decision to publish, or preparation of the manuscript.

**Competing interests:** The authors have declared that no competing interests exist.

**Abbreviations:** BLc, basolateral complex; cIN, cortical immature neuron; DCX, doublecortin; IN, immature neuron; NG2, Neuron Glia Antigen 2; OPC, oligodendrocyte progenitor cell; scINs, subcortical immature neurons.

generated neuronal cells that can remain in a state of arrested maturation for a long time [1–7]. These cells do not divide in postnatal or adult life, yet they express the same neuronal markers of immaturity generated by stem cell division in the canonical neurogenic sites [1,8,9], e.g., the cytoskeletal protein doublecortin (DCX; [10,11]), and the polysialylated form of the neural cell adhesion molecule (PSA-NCAM; [5, 9]). These markers can be expressed for a substantial duration before the cells restart maturation to functionally integrate into neural circuits, in a sort of "neurogenesis without division" [12–14]. These kinds of neurons have been characterized in most detail in layer II of the piriform cortex (cortical immature neurons; cINs), by following their progressive maturation in a DCX-Cre-ERT2/Flox-EGFP transgenic mouse model [6,8,12]. Remarkable interspecies differences of the cINs have also been revealed by systematic quantitative analysis carried out in the entire cerebral cortex of widely different mammals [15]. A far higher cell density of cINs is present in large-brained, gyrencephalic species with respect to small-brained, lissencephalic ones, thus suggesting that neurons in arrested maturation might have been increased across evolution in certain lineages to support cognitive functions in mammals with reduced stem cell-driven neurogenesis [16,17]. Accordingly, thousands of DCX⁺ immature neurons are present in cortical layer II of humans, extending into their widely expanded neocortex [18–20].

Over the years, the existence of neurons expressing DCX and PSA-NCAM has been repeatedly reported in subcortical brain regions, including amygdala, claustrum, and white matter tracts [21–32]. Due to their shared expression of immaturity markers with newborn neuroblasts [9,32], these cell populations were sometimes considered to be a possible product of noncanonical (parenchymal) neurogenesis (see, for example, [21,27]), though definitive evidence for their active division was elusive [26,29,30,32–34]. Several reports on DCX⁺ neurons in amygdala have included non-rodent mammals, suggesting the existence of interspecies variation [24,26,29,30]. A recent study revealed a small population of prenatally generated DCX⁺ cells in the mouse amygdala, mostly restricted to a very thin, previously undetected, paralaminar nucleus [7]. The same research group also described the occurrence of high numbers of similar cells in the paralaminar nucleus of humans [30]. These data suggest that the occurrence of subcortical immature neurons (scINs) like those in the cerebral cortex might be a conserved trait, yet with interspecies differences.

At present, the picture is fragmentary, due to the novelty of the field and because of different methods employed by researchers in the study of single species, often leading to divergent interpretations of the results [32,35]. On this basis, we undertook a systematic, interspecies investigation of the amygdala of diverse mammals by using the same approach employed for assessing the phylogenetic variation of cINs [15]. Eight mammalian species that widely differ in neuroanatomy (brain size, gyrencephaly, and encephalization) and other life history and socioecological features (life span, habitat, and food habit) were considered (Fig 1). They include two rodents characterized by short and long lifespans (mouse and naked mole rat, respectively), a small platyrrhine monkey primate (marmoset) and a large hominoid primate (chimpanzee), a very large brain-sized herbivorous perissodactyl (horse), a carnivoran

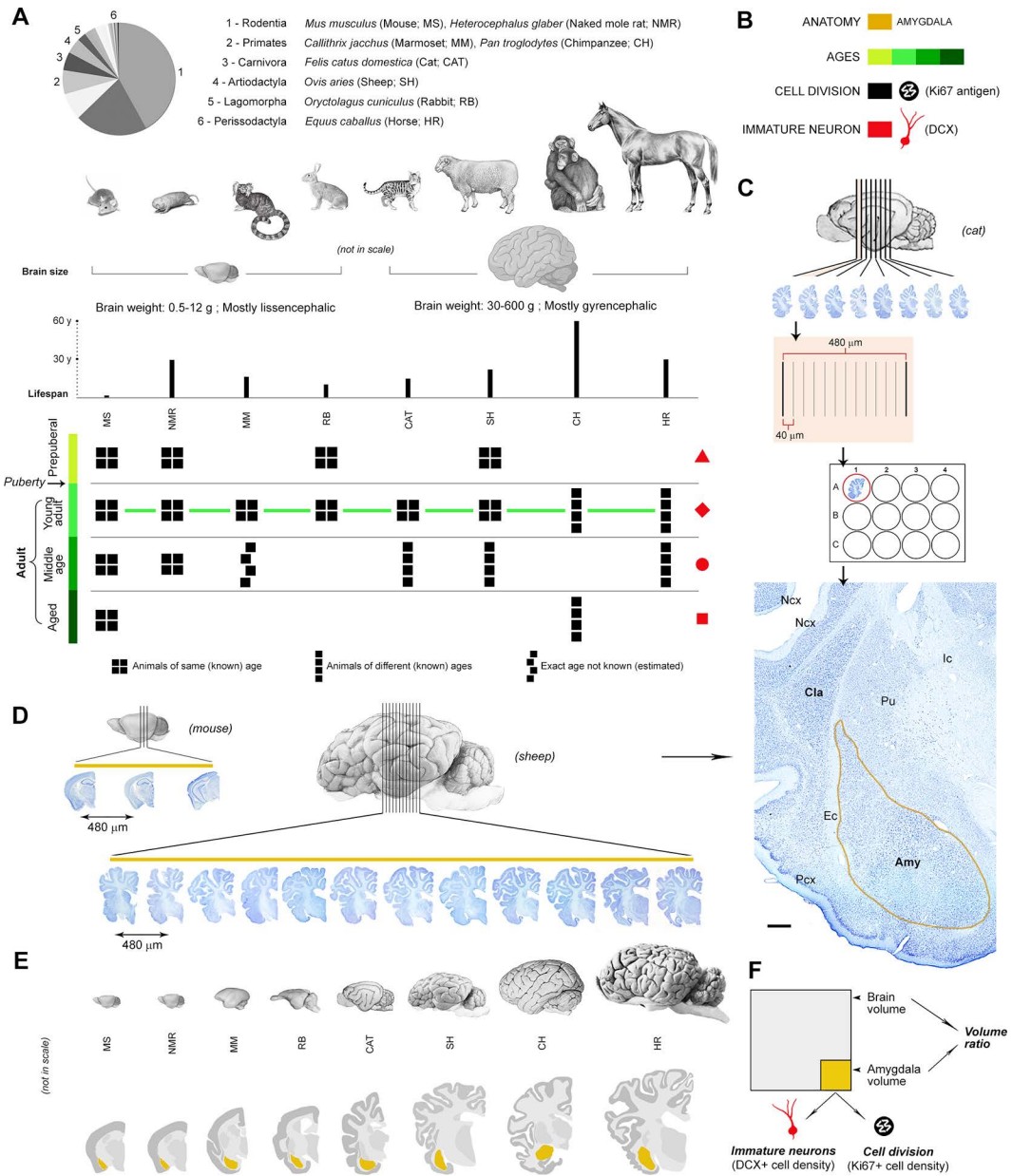

**Fig 1. Sample of species, ages, and brain regions of the mammals used in this study (further information in S1 Table; animal species are arranged from left to right according to their brain size).** (**A**, top) Mammalian species and orders (scientific name, common name – used hereafter – and abbreviation) with special reference to their brain size and life span. The pie chart illustrates the distribution of extant and recently extinct mammal species across orders according to the total number of recognized species (based on Wilson and Reeder; [36]; numbers indicate the position of the 6 orders considered here. (**A**, bottom) Different ages considered for each species, from prepuberal to old aged (see S1 Table for more detail on ages); all groups are composed of four individuals (black squares), and all species are available at the young adult stage (green line). Red shapes on the right are symbols to refer to age groups in the following figures. (**B**) Color code. (**C, D**) Brain tissue processing adopted to obtain comparable data in all species (mouse, sheep, and cat are represented as an example); serial coronal sections 40 µm thick of the entire hemisphere of each animal species were placed in multiwell plates to have an interval of 480 µm in each well to analyze the anterior-posterior extension of amygdala, followed by staining of sections and segmentation of the subcortical region based on histology (**C**, bottom); the final drawings of neuroanatomy in coronal sections are represented in **E**). (**D**) Different numbers of sections were obtained in each species depending on the brain size and consequent extension of the amygdala (see also S2 Fig and S4 Table). (**F**) By using the comparable method described above, volumes of the amygdala and whole hemisphere were calculated in each species. Then, counting of DCX$^+$ cells (immature-appearing neurons) and Ki67$^+$ nuclei (dividing cells) was performed in the amygdala (see S2 Fig). Created with brain icons from https://app.biorender.com/ and images reproduced with permission from Ref. [15]; this article is distributed under the terms of the Creative Commons Attribution License, which permits unrestricted use and redistribution provided that the original author and source are credited. Scale bar: 1,000 µm.

species showing the highest IN density in the cerebral cortex (cat; [15]), the rabbit, whose cIN density is intermediate between small-brained and large-brained mammals, and the sheep, as a gyrencephalic artiodactyl species in which INs of the amygdala where shown to be prenatally generated [29]. Different age groups were also investigated, reaching a total of 80 brains analyzed (Fig 1 and S1 Table). This study addresses open questions concerning: i) whether these cells in the amygdala share common features of nondividing "immature" neurons across species; ii) to what extent they display phylogenetic variation; iii) whether their amount varies with age and how this differs among species, and iv) whether they show similarities with their counterpart in the cerebral cortex. DCX+ (immature) neurons and Ki67+ (dividing) cells were studied to describe their occurrence, morphological and phenotypic types, density, topographical-topological distribution within the amygdala main subdivisions, and analyzed in a phylogenetic perspective.

## Results

### Amygdala subdivisions in the comparative sample of mammal species

To define the subregions of the amygdala across the mammalian species in our sample, we first established the topology and orientation of the amygdaloid complex along its full anterior-posterior extent, including its anatomical relationships with surrounding structures (Fig 2, left). Subnuclei were then identified by comparing histologically stained serial coronal sections (see Materials and methods) with existing neuroanatomical atlases and comparative studies from the literature (see Figs 2, S1, and References). Given the variability in how precisely amygdala subnuclei are defined across different species, and considering our goal of quantifying DCX+ cells and associating them with major subregions (see Materials and methods), we employed a standardized and slightly simplified scheme of subnuclei delineation. This scheme allows for consistent comparisons across species and focuses on three rostrocaudal levels of the amygdala: anterior, middle, and posterior (Fig 2, right; S1B Fig). For all species, we grouped subnuclei into three major divisions: the basolateral complex (comprising the lateral, basal, and accessory basal nuclei), the centro-medial complex (including the central and medial nuclei), and the cortical nucleus (Fig 2, right; S1 Fig).

### Characterization of the amygdala DCX+ neuronal population across mammals

The entire amygdala of eight mammalian species was examined with immunodetection of a panel of antigens (S2 Table). After testing immunocytochemical staining for the best-performing antibodies to detect DCX and Ki67 antigens, the polyclonal goat anti-DCX antibody from Santa Cruz and the polyclonal rabbit anti-Ki67 antibody from Abcam were selected ([19]; S2 Table). Despite slight differences in postmortem interval, fixation procedure, and fixation time among species (S1 Table), no substantial variation was observed in the quality and intensity of the staining (Fig 3). In addition to the amygdala as a region of interest, other regions from the same brains were used as internal controls, including the canonical neurogenic sites (forebrain subventricular zone, SVZ, and hippocampal subgranular zone, SGZ; Fig 4B) as a positive control for cell division, and the cerebral cortex (neocortex and paleocortex; Fig 4B), as a positive control for cINs and negative control for their possible cell division (absence of DCX/Ki67 co-expression).

DCX+ cells were found in the amygdala of all mammals studied, and at all ages in the sample, yet with evident interspecies differences in their density and distribution. Most strikingly, only a few, scattered DCX+ cells were detectable in adult mice (being entirely absent in some of the coronal sections analyzed), while extended networks of DCX+ cells were present in the amygdala of primates (Fig 3).

Two morphological DCX+ cell types previously reported in cortical layer II (cINs of both paleo and neocortex; [15,29] Fig 3A, 3C) were consistently detected: type 1, unipolar or bipolar, with a small cell soma (cell soma diameter range: 3–9 μm), and type 2, characterized by ramified dendrites and a larger cell soma (9–19 μm). Type 1 cells were very similar to the correspondent type in the cortex, while type 2 cells were split into two subtypes distinctive to the amygdala: type 2a (bipolar, with a ramified dendrite; soma size range 9–12 μm) and type 2b (showing multiple, ramified dendrites resulting

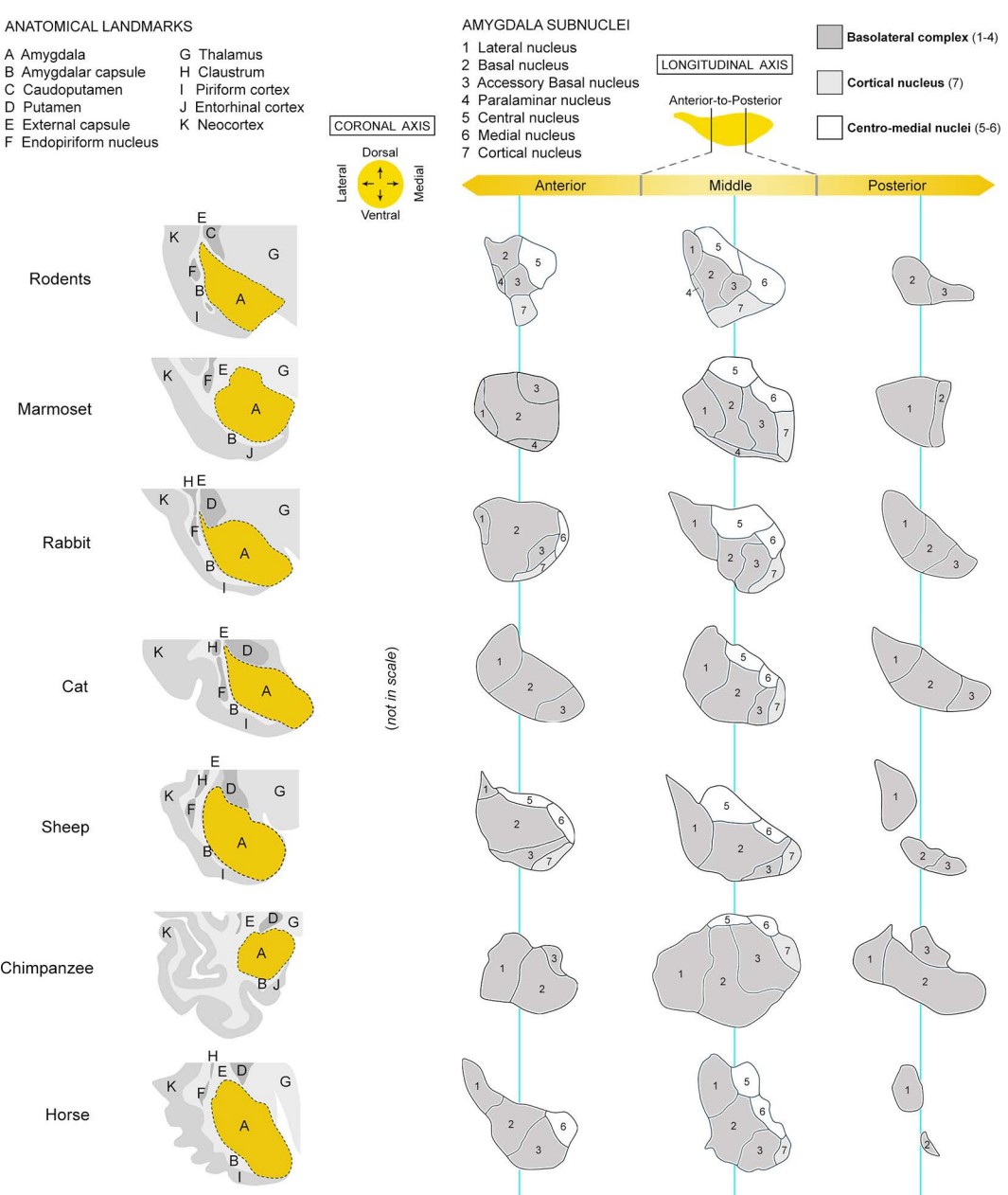

**Fig 2. Comparative neuroanatomy of amygdala in all mammals in this study: topology, surrounding structures, and main subnuclei segmentation.** On the left, the amygdala (orange) is represented with its surrounding structures, which can vary depending on the species' brain size and gyrencephaly (e.g., the amygdalar capsule of white matter is lateral in the amygdala of a mouse, becoming ventral or even ventromedial in primates). For subnuclei segmentation (right), the entire amygdala anterior-posterior length of each species (from 1.4 to 9.6 mm) was analyzed at 480 µm apart (one serial coronal section out of 12 stained with toluidine blue; cresyl violet in chimpanzees; from 3 sections in rodents to 20 sections in horse; see S2 Fig) and matched with atlases and comparative studies available in the literature: ([7,37,38] mouse; [39] naked mole rat; [40,41] rabbit; [28,42] cat; [43,44] sheep; [45–47] marmoset; [47,48] chimpanzee; [49], horse). The representation of subnuclei is shown in three parts of the amygdala (anterior, middle, posterior; different nuclei having different lengths) and topologically oriented according to the internal axes (see drawings of a representative middle section on the left). A simplified subdivision common to the eight species has been adopted (see S1 Fig) that was used for DCX+ and Ki67+ cell counting. The paralaminar nucleus has been represented only in species where it has been previously described (mouse and marmoset). Nuclei of the basolateral complex (BLc, including the paralaminar nucleus) are colored in dark gray. This interspecies mini-atlas was used to establish the topological and topographical distribution of the DCX+ and Ki67+ cells in both longitudinal (anterior-to-posterior extension, shown in Fig 6A) and coronal axes (lateral-medial and dorsal-ventral position, Fig 7). Animal species are arranged from top to bottom according to increasing brain size.

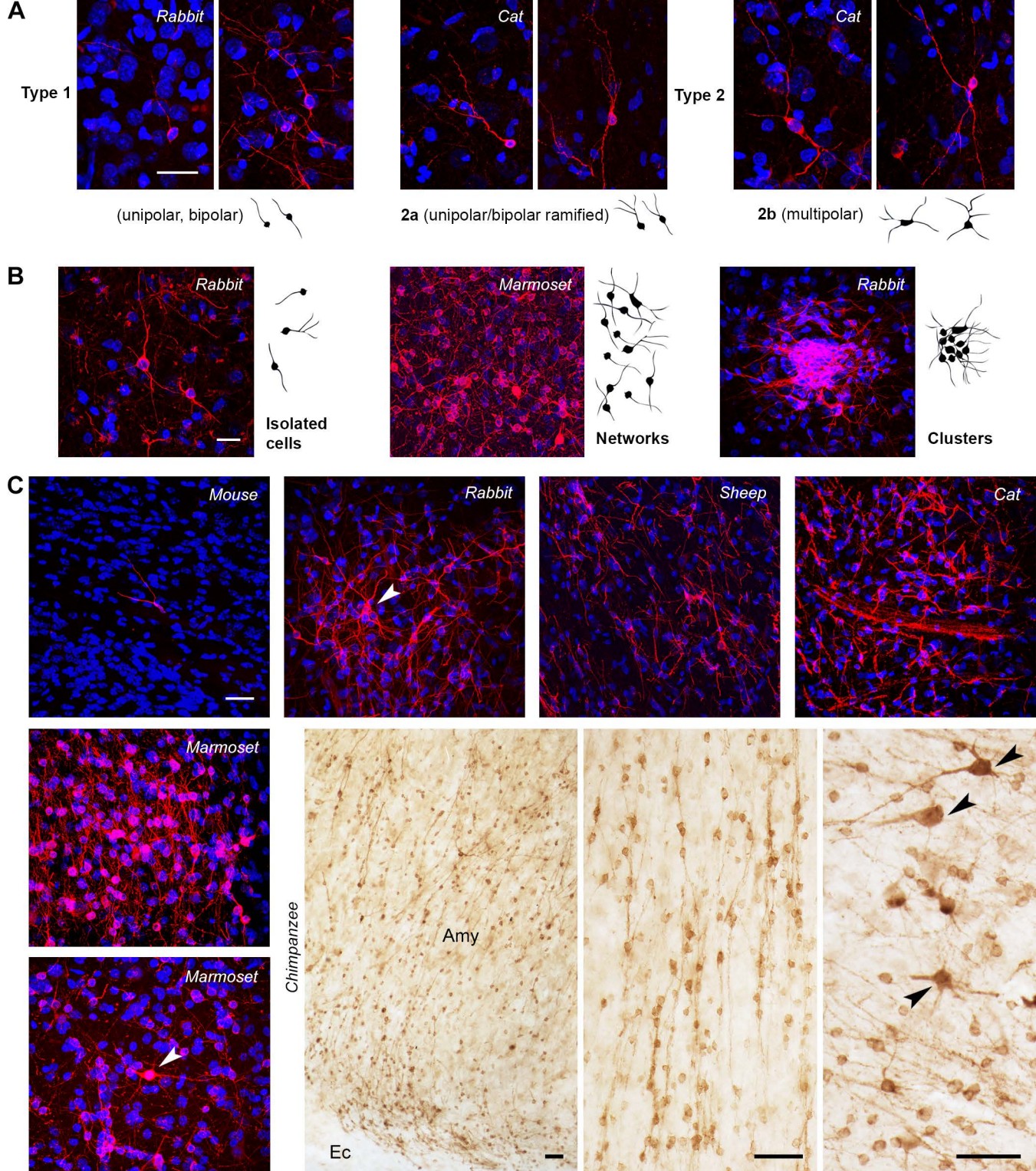

**Fig 3. Occurrence, morphology, and general distribution of DCX+ cells in the amygdala of mammals.** Red (immunofluorescence) and brown (diaminobenzidine staining; DAB), DCX; blue, DAPI. **(A)** In all species, the morphological cell types of DCX⁺ cells were reminiscent of type 1 (very simple morphology) and type 2 (complex cells) immature or dormant neurons described in the cortex [8,15]. **(B)** At least three types of cell distribution/

aggregation were observed, from isolated cells to tightly packed clusters. **(C)** Confocal fields of DCX⁺ cells in the amygdala of different mammals after immunofluorescence staining (except for chimpanzee specimens, photographed in light microscopy after DAB staining). Note the presence of scattered, isolated immunoreactive cells in mice with respect to dense, extensive networks in primates (the number of DCX⁺ cells appeared particularly high in chimpanzees; see internal controls in S3 Fig). Arrowheads indicate some type 2b cells, whose spatial distribution was random. All photographs come from the basolateral complex. Amy, amygdala; Ec, external capsule (amygdalar capsule); arrowheads, examples of type 2b cells. Scale bars: A–C (confocal), 30 μm; C (DAB), 50 μm.

in a multipolar morphology, and 12–19 μm soma size range; Figs 3A and S2D). In cortical layer II, type 1 and 2 cells have been recognized to represent different stages of maturation in a process leading the small (highly immature) neurons to become larger and more ramified (complex cells) after they restart the maturation process to become principal neurons of that layer (pyramidal neurons; [6,8,12]). Accordingly, the large, type 2b cells described here resemble the principal cell type of the amygdala (pyramidal-like Class I projection neurons; [50,51]), and their glutamatergic identity was confirmed by co-expression with the excitatory neuron transcription factor T-box brain 1 (Tbr1; [52,53] Fig 4A; higher magnification and separate channels in S4 Fig). The spatial distribution of the type 2 cells among the rich network of small, type 1 cells, appeared to be random. A small subpopulation of the DCX⁺ cells (percentages given in Fig 5C'), consisting of large, complex cells, mostly type 2b cells, also co-expressed NeuN (Fig 4A; tabulated data can be found in S1 Data), indicating they have started maturation. On the other hand, most of the DCX⁺ cells, including all type 1 cells and the remaining Type 2 cells, consistently co-expressed the "low adhesive" adhesion molecule PSA-NCAM on their soma and/or cell processes (Fig 4A), thus confirming their immaturity state.

Overall, the DCX⁺ cell types of the amygdala were reminiscent of those described in the cortex [6,8,12], thus suggesting that populations of immature neurons characterized by similar, progressive stages of maturation are likely shared by cerebral cortex and amygdala, their final product being tailored to the specific neuronal types of the two regions. These features prompted us to identify these cells to be neurons in arrested maturation (INs). On this basis, we sought to determine whether some of the DCX⁺ cells in the amygdala might be actively dividing, by performing DCX/Ki67 antigen double staining for confocal analysis to search for possible marker co-expression in all the species considered (S2C Fig). After counting a total of 856 Ki67⁺ nuclei in 252 confocal fields belonging to all species studied (aside from chimpanzees, not investigated in confocal microscopy), no co-expressions with DCX⁺ cells was ever detected, indicating that the DCX⁺ immature neurons and the proliferating cells in the amygdala most likely belong to distinct cell populations, as previously found in the cerebral cortex [15]. Based on this result, and on the frequent appearance of the Ki67⁺ nuclei as "doublets" displaying the typical characteristic of dividing oligodendrocyte progenitor cells (OPCs; Fig 4C) [54], the possible co-expression of the cell division marker with glial markers was investigated. Although NG2 (Neuron Glia Antigen 2) is considered the ideal molecular marker for the identification of early developing OPCs in rodents, its expression cannot be easily detected in postmortem tissues, which are usually too heavily fixed [55]. Accordingly, the marker worked well in perfused mice brains (see [56]) while giving poor results in most other mammalian brain tissues. As an alternative, we employed the pan-oligodendrocyte transcription factors SOX10 [57] and Olig2 [58] in double staining experiments against Ki67 antigen. Indeed, co-expressing cells were frequently observed in all the specimens examined (Fig 4C), thus confirming the glial nature of most dividing cells in the amygdala.

### Quantification of DCX⁺ neurons and dividing cells (cell densities) in the amygdala of different mammals

The number of DCX⁺ cells per mm² of amygdala was assessed through measurements using Neurolucida software in serial coronal sections spaced 480 μm apart, by counting all the cells in the amygdala segmented area (cell density/area; S2C Fig). Due to the different lengths of the structure in each animal species (from 1.4 mm in rodents to 9.6 mm in horses), different numbers of sections were analyzed (from 3 to 20; S2A Fig and S4 Table). This approach allowed us to

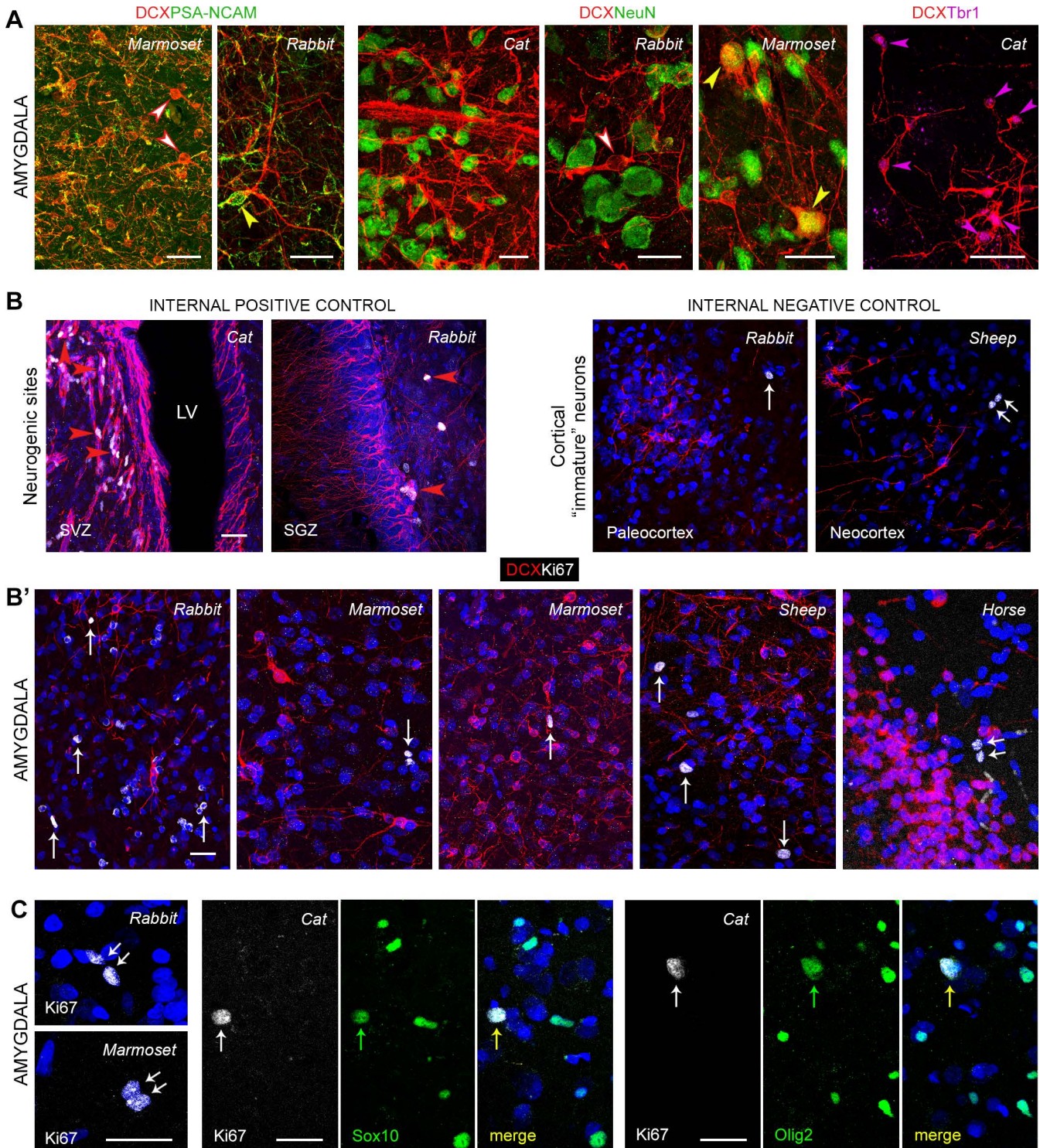

**Fig 4. Confocal analysis of different markers in the amygdala of mammals. (A)** Markers of immaturity (DCX and PSA-NCAM, left) are widely distributed and co-expressed in the scIN population; some DCX⁺ type 2 cells are devoid of PSA-NCAM (red and white arrowheads), while others still express it (yellow arrowhead). Most DCX⁺ cells, including type 2 cells (red and white arrowheads), do not express the marker for postmitotic neurons that start differentiation NeuN (middle); only a subpopulation of type 2 cells express NeuN, indicating they started the maturation process (yellow arrow-heads). Most DCX⁺ cells also co-express the marker for glutamatergic neurons Tbr1 (right; an enlargement with separate channels for each marker

is shown in S4 Fig). **(B)** Positive (neurogenic sites: subventricular zone, SVZ, and subgranular zone, SGZ; LV, lateral ventricle; red arrowheads) and negative (cerebral cortex) controls for detection of cell division. **(B')** Representative images of DCX/Ki67 antigen double staining showing total absence of co-expression in amygdala of any of the species considered. Note the occurrence of "doublets" (double arrow) in the cortical and amygdalar parenchyma. **(C)** Ki67 antigen staining in amygdala frequently revealed "doublets" (left) and was usually associated with oligodendrocyte progenitor cell division; accordingly, frequent co-expression was detectable in double staining of Ki67 antigen with the glial markers SOX10 and Olig2. All photographs in the amygdala come from the basolateral complex. Scale bars: 30 μm.

obtain a comparable value in all species, whatever the size of amygdala and the spatial arrangement of the DCX+ cells within the region. Given the nonhomogeneous distribution of the latter (see below), a direct cell counting was favored over a stereological approach, and to make the analysis as close as possible (comparable) to that previously performed in the cerebral cortex [15].

In Fig 5A, the results for quantifications in young adult animals are reported from all the species in this study (indicated by the green line in Fig 1). All nonrodent species (marmoset, rabbit, cat, sheep, chimpanzee, horse) showed a higher density of DCX+ cells than rodents (mouse, naked mole rat). Both primates (marmoset, chimpanzee) had significantly greater density of DCX+ cells (nonparametric Kruskal-Wallis test, $p < 0.05$), with a two orders of magnitude difference between mouse and chimpanzee (Fig 5A; tabulated data can be found in S1 Data). Differential counting of type 1 and type 2 cells was also performed (percentages reported in pie charts of Fig 5C). The small, highly immature cells were far more numerous (usually exceeding 90%) than the large, ramified cells.

The same analysis was carried out on Ki67+ nuclei. In comparison with data obtained for the DCX+ cell population, the density of Ki67+ dividing cells showed a quite different pattern, generally at low densities in all species (Fig 5B). Within this range, the rate of cell division was higher in rodents and more sporadic in nonrodent species, with a significant difference in mouse with respect to chimpanzee (nonparametric Kruskal-Wallis test, $p < 0.05$; Fig 5B).

The reconstruction of evolutionary change in the values of DCX+ cell density, DCX+ cells as a percentage of the basolateral complex of the amygdala, and Ki67+ cell density is shown in Fig 5E. Increased density of DCX+ cells occurred along the primate lineage, whereas an increase of Ki67+ cells occurred in a separate branch of the phylogeny that includes rodents and rabbits.

Overall, rodents displayed low DCX+ cell density and higher Ki67+ cell density (see S5 and S6 Tables for estimation of total INs and total dividing cells), while this pattern was inversed in primates, a general pattern further supporting the idea that INs and cell divisions in amygdala represent different cell populations. Also, the general distribution of the Ki67+ nuclei and DCX+ cells appeared quite different in all species considered, with the former randomly distributed in the entire coronal plane and the latter mostly grouped in a defined portion of the amygdala (Fig 5D), prompting further investigation of the spatial distribution of the two cell populations.

## Topological distribution of DCX+ neurons and Ki67+ dividing cells in the amygdala of mammals

Most DCX+ cells appeared as isolated and rare elements in rodents (mouse, naked mole rat), while frequently organized to form clusters and complex networks in other species (Fig 3B). The topological distribution of the cells in the amygdala was analyzed both in the entire longitudinal, anterior-to-posterior axis (Figs 6A and S5) and in the coronal plane (medial-to-lateral/ventral-to-dorsal axes; Fig 7), in all species. The anterior-to-posterior distribution is represented by histograms reporting the mean total cell number in each coronal section for each species (for both DCX and diving cells, S5 Fig) and the corresponding line plot for DCX+ cells including all species (Fig 6A). Marked interspecies variation was observed, with the DCX+ cells concentrated in different amygdala compartments of different species (e.g., mostly posterior in cat and sheep, while mostly in the middle in horse and primates).

By contrast, the distribution of the cells in the coronal plane was consistent across species, being prevalent in ventral, lateral or medial directions (mainly close to the amygdalar capsule), depending on the topological orientation of the

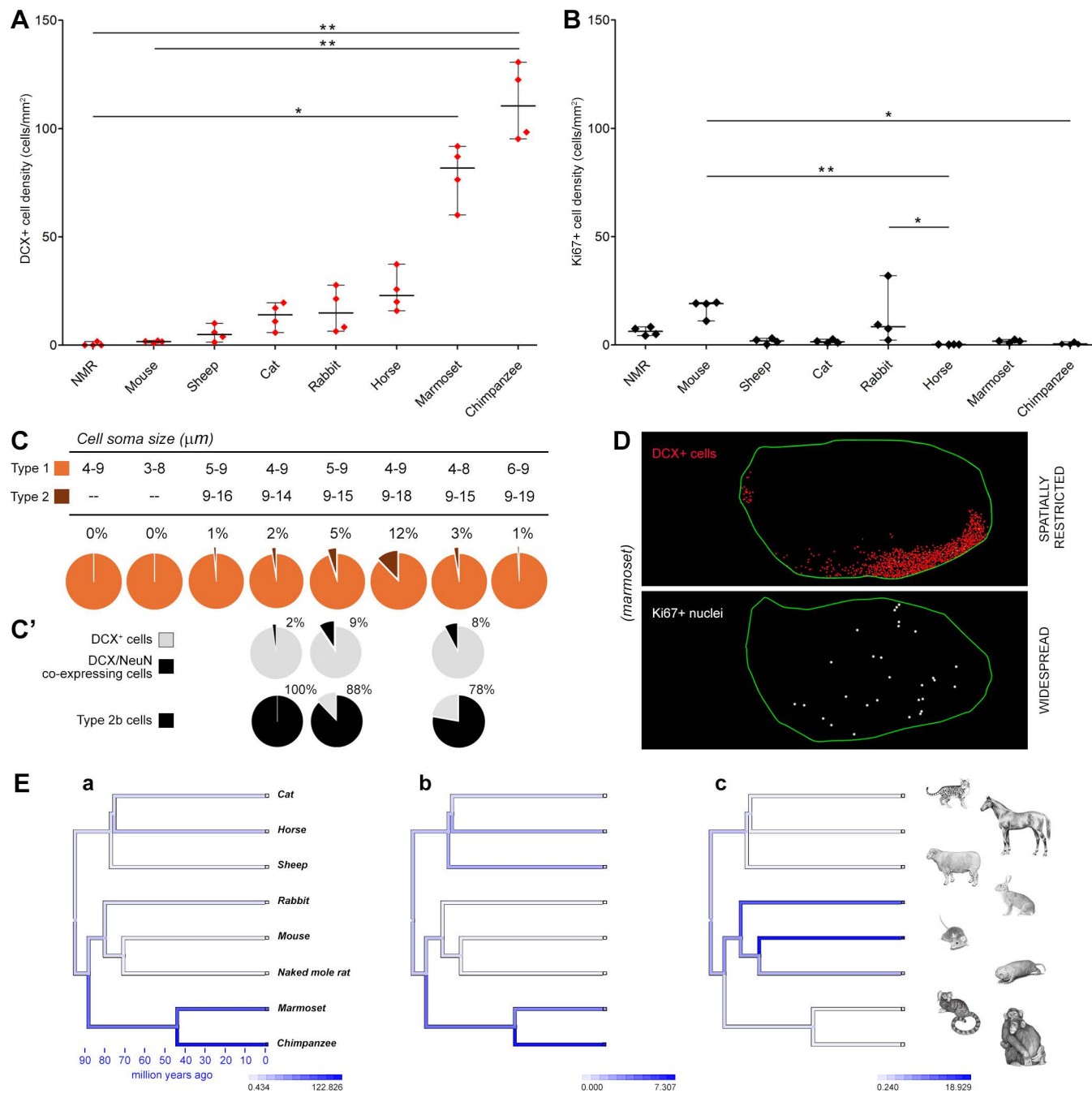

**Fig 5. Quantification of DCX+ neurons and Ki67+ nuclei in the amygdala of young adult mammals. (A)** Cell density and statistical analysis of DCX+ cells in the amygdala of eight mammalian species (listed in ascending order from left to right; see also the line plot with more extended scale in Fig 6A). A high degree of heterogeneity is detectable from rodents to primates, the latter showing the higher amount. **(B)** Cell density and statistical analysis of dividing cells in the amygdala of the same animal species and age. A rather minimal homogeneity is detectable, with slight prevalence in rodents and rabbits, and very low levels in primates. Note the sharp contrast between DCX+ neuron abundance (**A**) and Ki67+ dividing cell scarcity (**B**) in primates; nonparametric Kruskal-Wallis test, *$p < 0.05$; **$p < 0.01$. Tabulated data can be found in S1 Data. **(C)** Counting of type 1 and type 2 cells (see Figs 3A and S2D); top, cell soma diameter ranges; bottom, percentages are shown in pie charts (numbers indicate the percentage of type 2 cells). **(C')** Percentages of DCX+ neurons co-expressing NeuN (cat, rabbit, and marmoset); all co-expressing elements were complex cells (type 2 cells), most of them being type 2b cells (see Fig 4A, and S1 Data). A–C, Mammal species are arranged from left to right according to increasing DCX+ cell density. **(D)** DCX+ neurons (mostly restricted to the basolateral complex of the amygdala) and Ki67+ dividing cells (widespread in the entire region) have different topographical distribution, suggesting they belong to different populations (example given on marmoset; for other species see Figs 6A and 7). **(E)** Ancestral character

state reconstructions of trait evolution for DCX+ cell density **(a)**, DCX+ cells as a percentage of the basolateral complex of the amygdala **(b)**, and Ki67+ cell density (c) mapped onto the phylogeny. Animal icons reproduced with permission from [15]; this article is distributed under the terms of the Creative Commons Attribution License, which permits unrestricted use and redistribution provided that the original author and source are credited.

amygdala (Figs 2 and 6B), thus indicating that it might be a conserved trait, with the immature cells being mainly associated with specific subnuclei. To examine this further, the different amygdala subnuclei were identified by combining our anatomical mapping with existing atlases and previous descriptions in each species (see Figs 2 and S1 for details). Due to heterogeneous knowledge and terminology existing in the literature concerning the amygdala subnuclei across mammals [48,50,59–62], we adopted a simplified grouping which considers the standard subdivisions of the basolateral complex (lateral, basal, and accessory basal nuclei), the centro-medial complex (medial and central nuclei), and the cortical nucleus (Figs 2 and S1). In previous reports of comparative neuroanatomy, the paralaminar nucleus has not been identified and characterized as a separate nucleus in all species, being often considered as part of the basal nucleus [59,61]; for this reason, and due to its cellular composition (see Discussion), it has been included in the basolateral complex, referred hereafter to as BLc.

By observing the DCX-immunostained sections, it appeared that most DCX+ neurons were concentrated in the area of the BLc (see Fig 6B). To allocate more precisely the immature cells to each of the amygdala subnuclei, the percentage of area occupied by the cells was measured (see Materials and methods, Fig 6C, and S3 Table). As shown in Fig 7B, most of the DCX+ cells were consistently located in the nuclei forming the BLc of each species (in rodents, a few INs are segregated in the small area adjacent to the amygdalar capsule and attributed to the paralaminar nucleus; see also Ref. [7]. Notably, the association of the INs with the BLc was independent from variation in their density or anterior-posterior distribution, again suggesting that it is a conserved trait. The percentage of area occupied by the INs in the BLc varied across the species, with the greatest amount in primates, in parallel with DCX+ cell density (S3 Table and Fig 7B).

Overall, these data demonstrate phylogenetic variation in amygdala INs, with a focus on species differences in the BLc.

No DCX+ cells, or only a negligible amount, were found in the medial and central nuclei of the amygdala in both rodent and nonrodent species, while a modest number of them were present in the cortical nucleus in horses and chimpanzees (S3 Table).

As shown above (Fig 5D), by comparing Ki67-stained and DCX-stained adjacent sections, the difference in the distribution of the two cell populations was evident, the former being randomly scattered in the entire amygdala while the latter was mostly grouped in its ventral part. This difference was even more evident quantitatively, both considering whole cell density medians (Fig 5A, 5B) and those representing single coronal sections (comparison between left and right histograms in S5 Fig). Hence, quantitative analyses combined with topographical location of the cells confirm that INs and dividing cells in the amygdala belong to distinct populations.

Although small, bipolar/unipolar DCX+ cells with a leading-like process are commonly detectable as the most immature forms of the neurons (even in the cortex; [8,12,15]), they can be reminiscent of a migratory morphology [30]. To examine this possibility, an additional analysis was carried out on the occurrence and direction of cells displaying such morphology; we checked for the existence of higher densities of these cells at specific locations and for specific direction of their processes (see Materials and methods). The results of this analysis (reported in S7 Fig) revealed that the cells with migratory morphology were extremely rare with respect to the total of DCX+ cells (especially in large-brained species) and randomly oriented.

### Quantification of DCX+ cells, dividing cells, and amygdala volumes at different ages

To investigate whether the amount of DCX+ and dividing cells changes during the life span, we analyzed brains from the young adult stage to other available ages, including middle age in mouse, naked mole rat, marmoset, cat, sheep, horse,

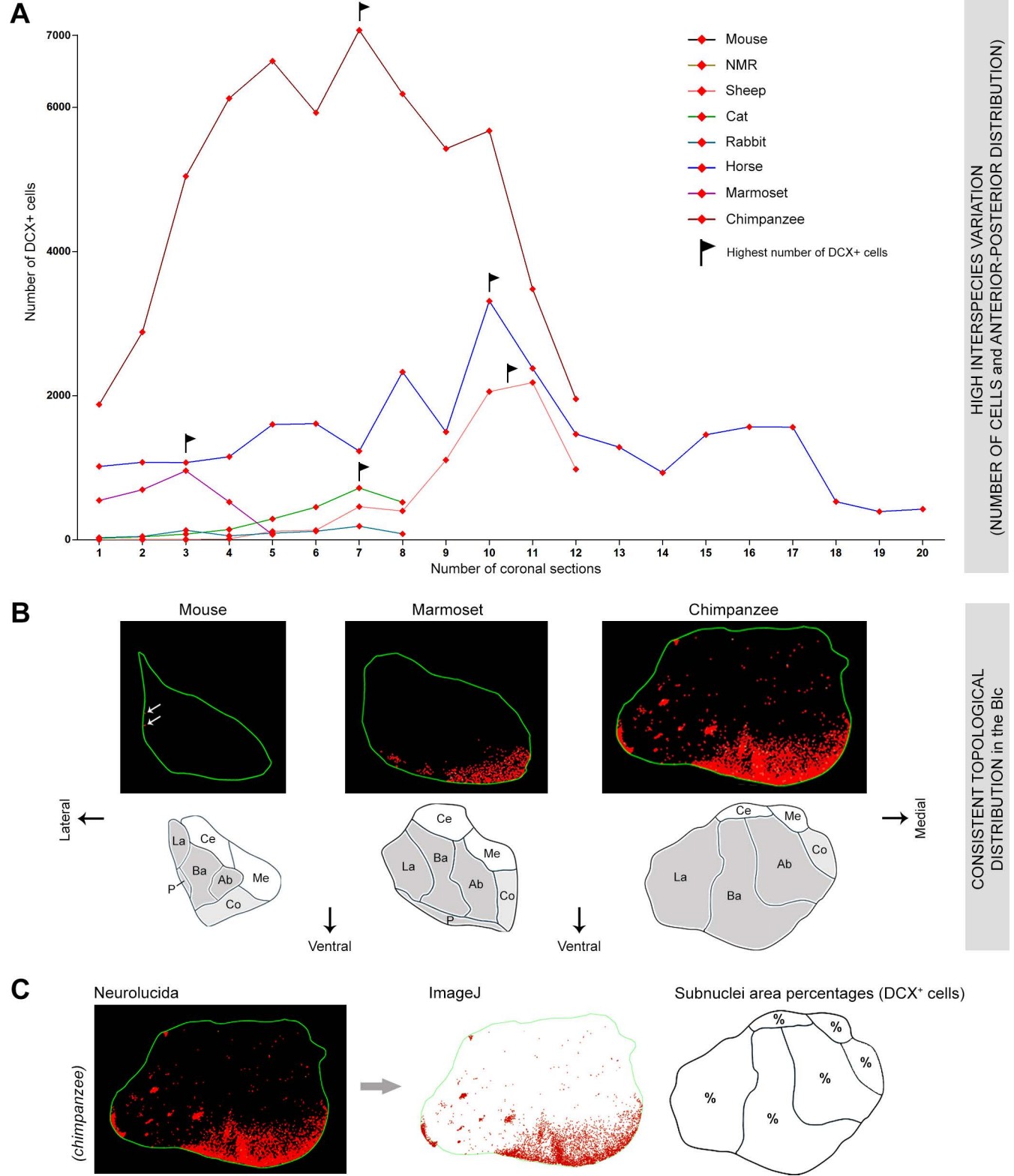

**Fig 6. Anterior-posterior and coronal (dorsal-ventral, lateral-medial) distribution of the DCX⁺ cells in the amygdala of mammals. (A)** Line plot indicating the mean total number of DCX⁺ cells counted in each coronal section in all species. Marked interspecies differences are present in the

anterior-posterior distribution: horses and primates show the highest amount in the middle, while cats and sheep have more in the posterior part. Flags indicate the highest number of cells in species showing the highest anterior-posterior differences (mostly gyrencephalic species and primates, indicated here; see S5 Fig for all species). **(B)** In contrast with the varying anterior-posterior distribution, when observed in the coronal plane, the DCX$^+$ cells of all species appeared consistently associated with the basolateral complex (dark gray area) despite its different topological orientation: from lateral in rodents, to ventral-lateral in marmosets, and ventral (even medial) in chimpanzees. This prompted a more detailed investigation of their distribution within different subnuclei (C). **(C)** To allocate the immature cells to each of the amygdala subnuclei, the DCX$^+$ cell counting markers were used (see Materials and methods and S3 Table; and Fig 7 for results).

and old age in mouse and chimpanzees (Figs 1 and 8). A prepuberal stage was also evaluated in mouse, naked mole rat, rabbit, and sheep.

A substantial drop (statistically significant in mice) was observed by comparing prepuberal with young adult stage (Figs 8 and S6). This fact is consistent with previous reports indicating that INs, like other forms of plasticity, mainly characterize juvenile ages, both in amygdala [7,30] and cerebral cortex [15]. Nevertheless, in large-brained gyrencephalic mammals (apart from horse) and primates, the age-related changes were not statistically significant, indicating a substantial stabilization in adulthood (Fig 8; nonparametric Mann-Whitney test, $p < 0.05$). In a previous report carried out in the mouse amygdala, the DCX$^+$ neurons have also been reported to drop in number very rapidly within the first two months of life [7], and we confirm here that only a few cells remain thereafter, being negligible at middle age and old stages (0.25 and 0.08 cells/section, respectively; see S5 Table and Fig 9). Even in another rodent species characterized by extended life span and retention of neotenic features, such as naked mole rats [63], no more DCX$^+$ cells were detectable from middle age onward (S5 Table). By contrast, in primates, particularly chimpanzees, the amount of scINs appeared rather unchanged, even between relatively distant life stages (e.g., young adult and old; Figs 8 and 9). By comparing mice and chimpanzees at these corresponding life stages, the DCX$^+$ neurons (total number estimation/hemisphere; see S5 Table) dropped from around 70 in young adults to 3 in aged stages (24-fold reduction) in the former, and from 700,000 to 470,000 (only half reduction) in the latter.

In parallel with the analysis of DCX$^+$ cell density, counting of Ki67$^+$ nuclei was also performed in all species and ages available. Again, a significant drop was observed in species analyzed at prepuberal stages (Figs 8B and S6). Particularly in rodents, which showed higher rates of cell division than other species, a significant drop occurs at juvenile stages. Subsequently, the levels of cell proliferation do not change significantly at later stages, remaining low in adulthood in rodents and nonrodent species (nonparametric Mann-Whitney test, $p < 0.05$; Fig 8B). Interestingly, rodents and primates again substantially differ from each other, the former having a higher amount of parenchymal cell division (still present at young adult stages) while the latter, along with other large-brained gyrencephalic species, displayed very low levels at all life stages (Fig 8B).

Finally, an analysis of amygdala volumes in different species at the same age (young adult stage) and at different ages was carried out, to measure both absolute volumes and amygdala/brain volume ratio (Fig 8C, 8D; tabulated data can be found in S1 Data). While the absolute amygdala volume, as expected, changed considerably with increasing brain size (from 1.9 mm$^3$ in mouse to 460 mm$^3$ in horse; Fig 8D, bottom), the amygdala/brain volume ratio was substantially similar in all species, thus independent from brain size (Fig 8D, top).

To explore the scaling relationship between DCX$^+$ and Ki67$^+$ cell densities with brain and amygdala volumes, least squares regression analyses were conducted (Fig 8E). For the relationship between DCX$^+$ cell densities with brain volume and amygdala volume, the results indicated a significant positive association (brain volume: $y = 0.419 + 0.479x$; $R^2 = 0.50$, $F(1.6) = 5.93$, $p = 0.051$; amygdala volume: $y = -0.661 + 0.467x$; $R^2 = 0.52$, $F(1.6) = 6.42$, $p = 0.044$.) In contrast, the relationship between Ki67$^+$ cell density with brain volume and amygdala volume displayed a significant negative association (brain volume: $y = 0.950 - 0.448x$; $R^2 = 0.72$, $F(1.6) = 15.24$, $p = 0.008$; amygdala volume: $y = 2.003 - 0.448x$; $R^2 = 0.79$, $F(1.6) = 22.13$, $p = 0.003$).

Fig 7. **Spatial distribution in the amygdala coronal plane. (A)** Topographical distribution of DCX⁺ cells within the amygdala of different mammals obtained from placing markers on cells in brain coronal sections used for cell counting (red dots: DCX⁺ cells; green line: amygdala perimeter). Top, the

anterior-to-posterior extension of the amygdala has been split into three parts (anterior, middle, and posterior; see also Figs 2 and S5) to compare the distribution in different coronal planes with that observed in the longitudinal axis. While the DCX+ cell distribution in the anterior-to-posterior extension of the amygdala appeared to be highly heterogeneous among species (Figs 6A and S5), that in the coronal plane was quite constant, being prevalent in the BLc regardless of its topological position (see Figs 2 and 6B). **(B)** Percentages of areas occupied by the DCX+ cells within the three main amygdala subdivisions (BLc, cortical nucleus, and centro-medial complex), indicating the prevalent association of the INs with nuclei of the basolateral complex (BLc, dark gray), and, to a lesser extent, with the cortical nucleus in large-brained species (horse and chimpanzee; light gray). The invasion of the BLc is particularly evident in primates, sheep, and horses; in addition, despite the representation of amygdala and subnuclei not being in scale, the relative volume of the BLc is far greater in primates than in rodents [45,48], thus making the percentages of areas occupied by the INs even higher. The very small percentages of areas occupied by the INs in rodents do correspond to their highly restricted location within the small paralaminar nucleus, without invading the BLc. Percentages for each of the subnuclei are reported in S3 Table. Mammal species are arranged from top to bottom according to increasing DCX+ cell density (the amount of cells in each field of view is not always representative, especially due to the BLc highly extended both anteriorly and posteriorly).

## Discussion

### DCX+ cells in the amygdala as neurons in arrested maturation

In the present study, at least 10 features commonly attributed to neurons in arrested maturation, or "immature neurons" (INs), previously described in the cerebral cortex [1,2,5,8] were consistently associated with the DCX+ neurons residing in the amygdala of widely different mammals, indicating the existence of scINs rather than newborn neurons (Table 1). In summary, these cells strictly fit with the morphological subtypes described for the progressively maturing neurons of cortical layer II, with a greater prevalence of small, simple-shaped cells and a minor fraction of complex morphologies reminiscent of the principal cell type of the region (pyramidal neurons in the cortex, Class I neurons in the amygdala). Most of the scINs co-express the markers of immaturity DCX and PSA-NCAM, while a small subpopulation of large and complex cells (type 2 cells, mostly type 2b) also express NeuN, an RNA-binding protein detectable in postmitotic neurons that have begun differentiation [64]; the scINs never co-express nuclear proteins associated with cell division, and display an utterly different spatial distribution with respect to proliferating cells, which are highly widespread and mostly identifiable with parenchymal glial elements known to represent the larger population of proliferative cells in the brain [54,65,66]. Also, the remarkable phylogenetic variation observed in the amygdala, with a large prevalence of scINs in primates compared to rodents, points to their nature as neurons in arrested maturation.

The existence of scINs is further supported by previous reports carried out in sheep and mice injected as embryos with 5′-bromo-2′-deoxyuridine, in which the DCX+ cells in the amygdala and claustrum of lambs [29] and those in the amygdala of mouse pups [7] were found to be prenatally generated, like the cINs. Though these experiments cannot be easily replicated in most large-brained, gyrencephalic species, our findings discussed above strongly indicate that populations of scINs exist in the amygdala of mammals. Overall, multiple lines of experimental evidence converge to define these cells as a reservoir of nondividing, young neuronal elements independent from stem cell-driven neurogenesis and reminiscent of their counterpart in the cortex [15].

**Interspecies variation in the amygdala of mammals.** In the present study, densities of DCX+ scINs were evaluated in a comparable way within the amygdala of 80 brains belonging to phylogenetically and neuroanatomically diverse mammals at various ages, by following the same approach previously employed to study the phylogenetic variation of cINs [15]. As a result, remarkable interspecies differences were found: the scINs were extremely rare in rodents and very abundant in large-brained gyrencephalic species, including perissodactyls, artiodactyls, carnivorans, and primates (Figs 5A and 6A). The density of scINs was also positively correlated with brain and amygdala volume (Fig 8E). This general pattern was reminiscent of that reported for cINs, whose densities were also related to brain size and gyrencephaly [15]. Additionally, beyond allometric scaling, the scINs seem to be more numerous especially in primates (which in our sample included both large-brained, gyrified chimpanzees and small-brained, mostly lissencephalic marmoset monkeys). While the INs are abundant in both cortex and amygdala of chimpanzees, few cINs were observed in the neocortex of

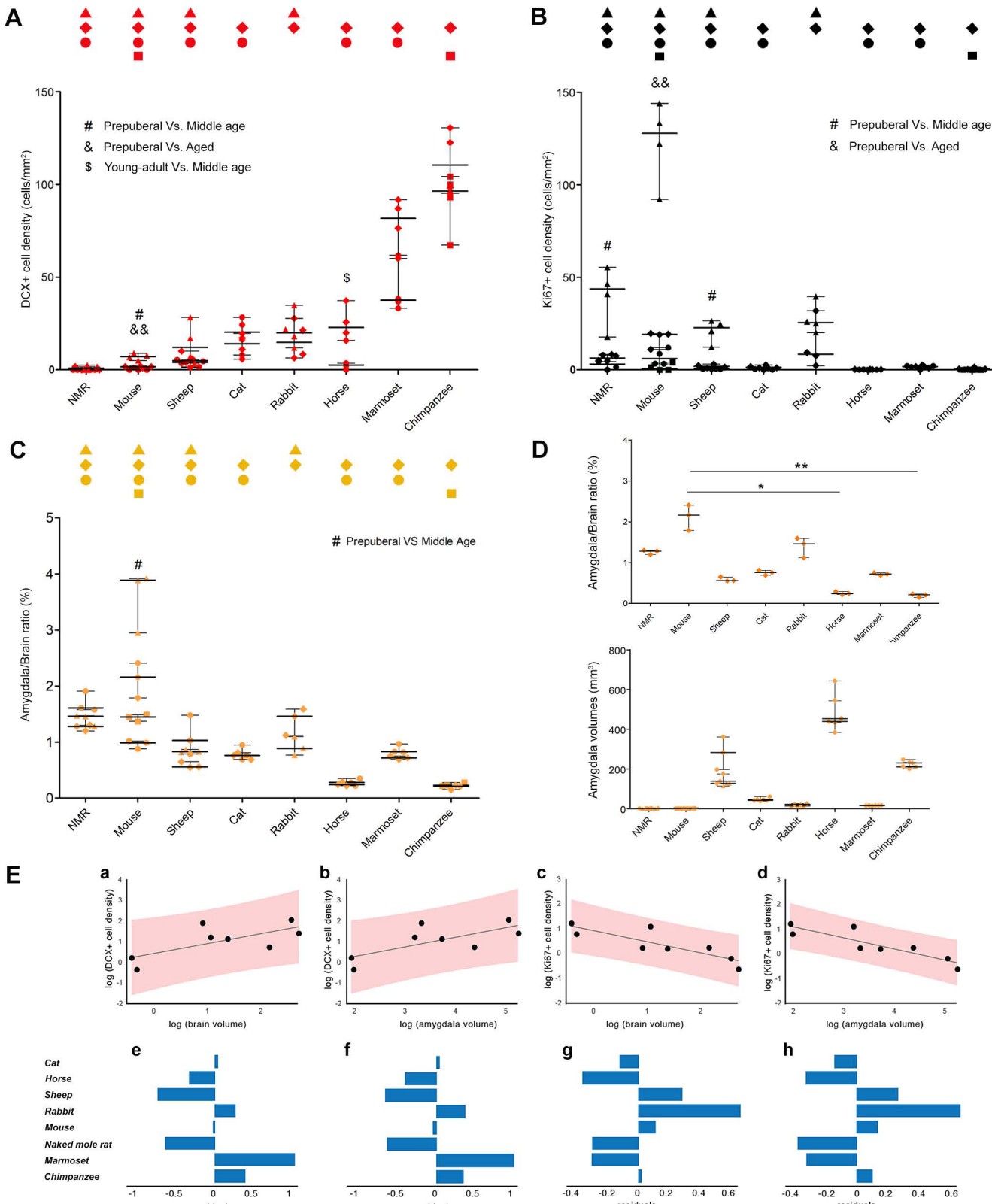

**Fig 8. Quantification of DCX+ neurons, Ki67+ nuclei, and amygdala volumes at different ages. (A)** Cell density and statistical analysis of DCX+ cells in the amygdala of eight mammalian species at different ages (all ages investigated are included; symbols indicating age groups are indicated in

Fig 1A). **(B)** Density and statistical analysis of dividing nuclei in the amygdala of eight animal species (same as in A) at different ages (all ages investigated are included). See enlargements of the A and B plots in S6 Fig, with color code for ages in addition to symbols. **(C, D)** Amygdala volume estimation. (C) Amygdala/brain volume ratio at all ages investigated; note the substantial invariance of volumetric ratio through different animal species and ages. (D) Amygdala/brain volume ratio at young adult age (top), and amygdala real volumes in each animal species at different ages (bottom); while the absolute volume of the subcortical region is higher in large-brained species (as expected), the volumetric ratio with respect to the whole brain is substantially stable, slightly lower in large-brained ones. A–D, Animal species are arranged from left to right according to increasing DCX$^+$ cell density; nonparametric Mann-Whitney test, $^{\#\&\$*}p < 0.05$; $^{\&\&**}p < 0.01$. **(E)** Least squares regression of DCX$^+$ cell density against brain volume (a) and amygdala volume (b), with associated residuals of the regressions shown underneath (e and f). Least squares regression of Ki67$^+$ cell density against brain volume (c) and amygdala volume (d), with associated residuals of the regressions shown underneath (g and h). All regression plots are on a log scale and show the 95% prediction intervals. Tabulated data can be found in S1 Data.

marmosets [15]. Future studies should examine a greater range of primate species that vary in brain size to confirm that the observed increase in amygdala scINs is consistent in this phylogenetic group.

In the cerebral cortex, such variation has been recently explained as a trade-off between different types of developmental processes involving stem cell-driven adult neurogenesis versus neurons in arrested maturation [17]. This might have occurred in brain evolution of mammals as a mechanism to assure the most appropriate type of plasticity is available, tailored for the brain size, life history characteristics, and behavioral adaptations in each species. Following this view, rodents have maintained a dependence on olfaction to navigate in their environments, while larger-brained species show specializations for experience-dependent learning related to neocortical expansion [67]. On this basis, the high number and widespread occurrence of INs found in the amygdala of primates might be linked to the increasing importance of this subcortical region in the integration of complex social interactions that can be crucial for their survival and reproductive success [26,61,68]. Primates share an increasing importance of some amygdala circuits and functions with respect to rodents [60,61], including their connections with the cortex [48], a fact that is reflected in amygdala subnuclei scaling (discussed below).

**Phylogenetic variation of amygdala matches subnuclei scaling.** While the general amygdala/brain volume ratio has remained relatively unchanged among widely different mammals ([48,60,69–73]; confirmed here by volume estimation in all species, see Fig 8C, 8D), some features of amygdala anatomy display phylogenetic variation, specifically in the scaling patterns of individual nuclei [48,59–61,71]. The lateral, basal, and accessory basal nuclei are proportionally larger in primates than in rodents, with respect to other nuclei (central, medial, and cortical), which differ relatively little [48,59–61]. These changes are also reflected in connectivity: the amygdala basolateral nuclei are broadly connected to the neocortex, while the centro-medial and cortical nuclei mostly communicate with brainstem, hypothalamus, autonomic system, and olfactory structures ([48, 59-61,72]). Accordingly, the increase in the basolateral complex volume has been correlated with increases of neocortical volume among primates, as a response to increased processing demands from the neocortex [69]. In line with the expansion of the associated cortical territories, the subnuclei of the BLc are preferentially expanded in primates compared with rodents (62% of total amygdala volume in macaques, and 69% in humans, with respect to 28% in rats; [48,61]); in addition, compared to rodents, the primate amygdala has lower neuronal density and larger neuropil volume, which have been associated with greater complexity in dendritic arborizations and axonal innervation, enabling a greater capacity to integrate information [48]. Hence, in contrast to the cortico-medial region, which is linked with olfaction and automatic, defensive reactions, the BLc has predominant connections with the neocortex ([61,62]; Fig 9).

Here we show that the remarkable increase in the number of scINs between rodents and primates (with other species being in an intermediate position), appears to be associated most closely with the BLc, or more generally with the cortico-basolateral amygdala, following its phylogenetic expansion (Fig 9). Accordingly, the scINs express the marker for pallial origin Tbr1, and the cortico-basolateral amygdala is developmentally a region of pallial origin particularly enriched in glutamatergic neurons (in contrast with the CM, mostly containing GABAergic neurons [74]). The phylogenetic variation of the scINs matches with the amygdala's expanding subnuclei and connectome, similar to what has been observed for cINs in

PLOS Biology

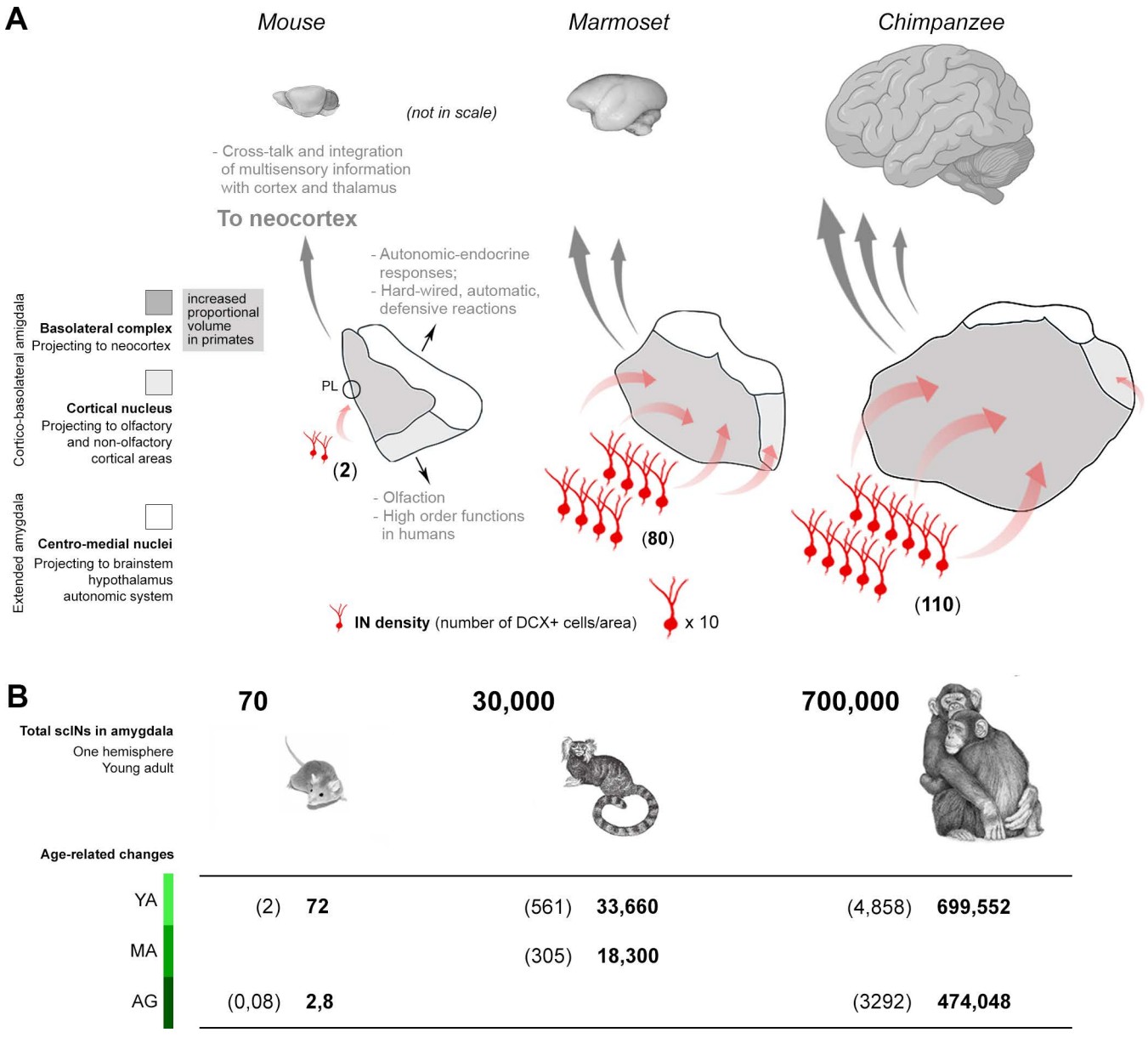

**Fig 9. Occurrence of subcortical immature neurons (scINs) in the amygdala of mouse and primates: relationship with evolution of subnuclei scaling and cortical connectomics. (A)**, The mammalian amygdala can be split into three main parts considering the basolateral complex (BLc; dark gray; here, including the paralaminar nucleus), the cortical nucleus (Co; light gray), and the centro-medial nuclei (CM; white). While CM and Co nuclei do not change substantially in their relative volume among animal species, the BLc is markedly larger in primates (up to 60%–70%) with respect to rodents (around 30%; here not entirely visible since not in scale and because the expansion is linked to the entire anterior-to-posterior volume of the region). Such an expansion has been related to increasing projections to the neocortex typical of primates (large gray arrows; [48,61]; see text). Despite a large difference in the number of scINs (red cells) found in primates with respect to rodents (see Figs 5A, 6A), as well as differences in the anterior-posterior distribution (see Figs 6A and S5), their prevalent occurrence within the basolateral complex is a conserved trait (some scINs were found in the cortical nucleus of horse and marmoset; to a lesser extent in chimpanzee; see Fig 7). While mostly restricted to a very small paralaminar nucleus (PL) in rodents, the scINs largely extend into the BLc in primates (red arrows and Fig 7B). **(B)** After scIN total amount estimation in the amygdala of each hemisphere (young adult age group, top), chimpanzees possess a reservoir of undifferentiated cells four orders of magnitude higher than mice, and this reservoir is maintained through ages in primates, while decreasing in rodents (bottom; quantitative data extracted from S5 Table). In brackets, average number of DCX+ cells in a coronal section of the amygdala; in bold, estimation of total number of DCX+ cells/hemisphere. YA, young adult; MA, middle age; AG, aged. Created with brain icons from https://app.biorender.com/ and images reproduced with permission from Ref. [15]; this article is distributed under the terms of the Creative Commons Attribution License, which permits unrestricted use and redistribution provided that the original author and source are credited.

**Table 1. Summary of the observations from the present study concerning features shared by different mammals indicating the DCX⁺ cells in the amygdala are "immature" neurons rather than newborn elements.**

| Features | Species | Figures |
|---|---|---|
| Expression and co-expression of typical markers of neuronal immaturity (DCX, PSA-NCAM) | All species | Figs 3 and 4A |
| Type 1 (small soma, unipolar/bipolar) and type 2 morphology (large soma, complex dendritic tree) typical of cINs and corresponding to different maturational stages [8,15] | All species | Fig 3A |
| Complex (type 2b) cells reminiscent of the principal neuronal type in amygdala | All species | Fig 3A |
| High percentage of type 1 cells and low percentage of type 2 cells | All species | Fig 5C |
| Low percentage of DCX⁺ cells (mostly type 2, complex cells) co-expressing the maturational marker NeuN | All species* | Fig 5C' |
| Different topographical distribution of DCX⁺ cells (grouped in restricted areas and associated with specific subnuclei) with respect to dividing cells (widespread), indicating they belong to distinct populations | All species | Figs 5 and 6 |
| Absence of co-expression with Ki67⁺ nuclei, indicating they are not dividing | All species* | Fig 4B' |
| High-rate co-expression between Ki67⁺ nuclei and glial markers (Olig2, SOX10), indicating that most dividing cells in the amygdala are glial elements | All species* | Fig 4C |
| Frequent detection of Ki67⁺ "doublets" indicating OPC division | All species | Fig 4C |
| Remarkable variation in number (cell density) between mammalian species, as previously described for cINs [15] | All species | Figs 5A and 6A |
| Substantial maintenance of the IN cell populations at adult and old stages in nonrodent species | Cat, Sheep, Chimpanzee, and Horse | Figs 8 and S6 |

*Chimpanzees were not tested for immunofluorescence double staining.

the expanded neocortex of gyrencephalic species [15]. Of importance, we show that in large-brained gyrencephalic species and primates, the occurrence of immature cells goes well beyond the paralaminar nucleus, an area that in most mammals is not classified as a subregion in its own, likely because it appears mostly composed of a mass of small, densely packed immature cells rather than having a specific histology. Although the fundamental function of the amygdala in fear and emotional learning is conserved across species, the region is under greater influence of cortical activity in primates, integrating contextual information linked to more complex behaviors such as social interaction and cognition [48,60,61,75]. Due to its extensive connections with much of the cortex, the amygdala is involved in the processing of salience, significance, ambiguity, and unpredictability, thus playing a role in selectively processing the inputs that are the most relevant to the goals of the animal [76]. Considering that the regions of high expansion are those that mature later in development to take advantage of experience and social learning [77], the greater occurrence of scINs may provide a substrate for plasticity in the form of a reservoir of undifferentiated cells subserving highly sophisticated functions.

**A reservoir of subcortical INs is maintained through the life span in primates.** The amount of DCX⁺ neurons and proliferating cells was investigated longitudinally in each species to examine possible age-related variation (Fig 1A). Results and statistical analysis are shown in Fig 8A. Despite a progressive age-related reduction observed in all species (significant only in mouse and horse, and generally at the transition from young to older ages), the number of INs was rather stable in adult primates. This pattern is particularly evident when comparing mice and chimpanzees, both studied at young adult and aged stages (Figs 8A and 9B): the chimpanzees, besides having far more abundant scINs, maintain many of them through adult and old ages, while they appear to be depleted very early in mice. Our results in mice are consistent with a recent report where DCX⁺ cells were investigated in the amygdala of young animals (P7 to P60; 7), showing a sharp decline through adolescence and reaching very low levels in the 2-month-olds. We extended the analysis to older ages and found only negligible numbers of DCX⁺ cells (0.25 and 0.08 cells/section in middle-aged and aged mice, with only 3 total cells in the entire amygdala of aged animals; S5 Table). By contrast, estimations of the total number of scINs per hemisphere in chimpanzees revealed a reservoir of 470,000 cells in aged individuals, which remained from the

nearly 700,000 of young adult animals (S5 Table and Fig 9B), suggesting a role for these undifferentiated cells through the life span. In accordance with this view, confocal analyses to examine co-expression of DCX and PSA-NCAM indicated that most scINs remain in a state of immaturity at adult and old ages, while the percentage of those displaying signs of starting maturation (co-expression of NeuN) was very low (around 2%–9% of all DCX+ cells, all corresponding to type 2, "complex" cells, mostly type 2b cells; Figs 4A and 5C'). Overall, the age-related trend of scINs is reminiscent of that described in the cerebral cortex, where the cINs also undergo a dramatic drop at young ages in mice (similarly to adult neurogenesis [56]) while remaining at higher levels in large-brained gyrencephalic species, especially chimpanzees [15,56]. Yet, a different ratio can be observed between type 1 and 2 cells in cortex and amygdala: the reservoir of highly immature cells (type 1 cells) is larger in amygdala (the type 2 cells being restricted to a range from 0% to 2%, except for horses; Fig 5C) than in the cortex (the cINs showing a range of 3–14%, reaching 44% in mice; see [15]). In a descriptive study performed on the human cerebral cortex at different ages spanning from neonatal to very old stages, many layer II DCX+ neurons (cINs) were still present in the neocortex of old individuals while the occurrence of DCX+ cells in the hippocampus of the same specimens was quite reduced [20].

Overall, our approach across mammal species and ages reveals that in nonrodent species (especially primates, and other large-brained, long-living species), the amygdala may rely on abundant populations of highly undifferentiated scINs, whose amount does not deplete sharply in juvenile ages, reaching a stabilization in adult and old individuals. A previous study performed in humans by Sorrells and colleagues [30] mainly focused on the drop in the number of DCX+ cells after adolescence. They also described the persistence of immature cells in adult and old individuals (up to 77 y of age), though their analysis was performed on a few adult/old human samples using tissue blocks from the temporal lobe, in which the amygdala was not considered in its entire extension in the coronal plane, their quantitative analysis being restricted to the paralaminar nucleus. The findings of our systematic study in two primate species, considering the whole volumetric extension of the amygdala, agree with that study in most respects, yet suggest that the human amygdala may host more scINs than currently thought.

The reason why such a great number of undifferentiated neurons persist through adulthood and old ages in the amygdala (and cortex) of large-brained, gyrencephalic species and primates is at present unknown. The idea that the immature cell reservoir is progressively depleted through young ages (e.g., integrated into preexisting circuits after resuming maturation) can only partially explain the current data. The scINs are maintained through life for some, at present unknown, reasons, likely linked to a need for plastic changes even at advanced ages. Our analysis of migratory-like cells (S7 Fig) showed that protracted cell migration within the amygdala is not a significant occurrence; small, unipolar/bipolar DCX+ cells correspond to the typical appearance of highly immature elements (even in the cortex, wherein the same DCX+ type 1 cells are known to not migrate [8,12,15]). As suggested by Sorrells and colleagues [30] in humans, it cannot be excluded that some scattered cells might retain a short-range migratory behavior to adjust their final location, yet this possibility would not explain the existence of large numbers of undifferentiated neurons.

Future studies are needed to explain the maintenance of an IN reservoir in nonrodent species and to investigate possible alternative fates and physiological roles of the INs in brain plasticity, functioning, and aging.

**Concluding remarks.** We showed that a population of young neurons (scINs) sharing features with those previously described in the cerebral cortex (cINs) is present in the amygdala of widely different mammals, displaying remarkable phylogenetic variation. Their density spans nearly two orders of magnitude from mouse to chimpanzee, and an estimation of their total amount in each hemisphere indicated a four-order magnitude difference, with 700,000 scINs in the 5.7 mm long chimpanzee amygdala (Fig 9 and S5 Table). By considering these numbers, together with their counterpart in the cortex (the cINs were estimated around 2 million cells/hemisphere in the chimpanzee [15]), the cINs and scINs are candidates for the most consistent reservoir of immature/plastic cells in primate brains, thus representing the only example of a substrate for structural plasticity in mammals that increases in association with brain size [78].

Overall, the scINs might represent a reserve of young cells tailored for cognitive functions in highly social mammals, supporting the idea that they may have been selected across evolution (along with the cINs) in a trade-off with stem

cell-driven neurogenesis, which is more active in short-living animal species, mostly relying on olfaction for their survival (e.g., rodents [16,17]). Notably, the current findings link the scINs to the expanding subnuclei of the amygdala's BLc, which are highly interconnected with the cerebral cortex, thus underlying the importance of cells in arrested maturation in amygdala-cortex interactions. Since emotion and social dysfunctions arise in psychiatric illnesses through altered connectivity between the amygdala and cortical regions [79], the reservoir of young neurons in arrested maturation existing in primate brains may have future potential in the prevention and therapy of neurodevelopmental disorders, possibly as a broad substrate for the brain reserve (or cognitive reserve) in aging [80,81]. These advances in our knowledge can facilitate the translation of research results obtained from different animal models, particularly laboratory rodents.

## Materials and methods

### Ethic statement

All experiments were conducted in accordance with current laws regulating experimentation in each country/institution providing the brain tissues (see below description for each group of animals).

Mice came from the Neuroscience Institute Cavalieri Ottolenghi (NICO) animal facility (Orbassano, Turin, Italy) and were treated under authorization of the Italian Ministry of Health, code RRID:MGI:3696370; courtesy of Serena Bovetti and Charles River Laboratories.

Naked mole rats were extracted at Queen Mary University of London (UK) a few minutes following euthanasia in accordance with Schedule 1 of the Animals (Scientific Procedures) Act 1986.

For rabbits, all experiments were in accordance with the European Communities Council Directive of 24 November 1986 (86–609 EU) and the Italian law for the care and use of experimental animals (DL.vo 116/92). All procedures carried out in this study were approved by the Italian Ministry of Health (authorization n. 66/99-A; 8 October 2009) [82,83].

Sheep, horse, cat, marmoset, and chimpanzee brains were extracted postmortem. Sheep (apart from prepuberal samples, see below) and horse samples originated from a commercial slaughterhouse and were treated according to the European Community Council directive (86/609/EEC) on animal welfare during the commercial slaughtering process and were constantly monitored under mandatory official veterinary medical care. For prepuberal sheep, animal care and experimental treatments complied with the guidelines of the French Ministry of Agriculture for animal experimentation and European regulations on animal experimentation (86/609/EEC) and were performed in accordance with the local animal regulation (authorization No. 006352 of the French Ministry of Agriculture in accordance with EEC directive) [84].

Cat brains were collected at the Department of Comparative Biomedicine and Food Science of the University of Padova; all specimens belong to animals undergoing diagnostic necroscopy, and owners signed a waiver authorizing the use of tissues for research.

Marmoset handling and tissue collection were conducted according to the Animal Welfare Act (AniWA) of the Federal Food Safety and Veterinary Office, Switzerland, and the Institutional Animal Care and Use Committee at The George Washington University, and follow the ethical guidelines outlined by the American Society of Primatologists.

Chimpanzee experiments were conducted following the international guiding principles for biomedical research involving animals developed by the Council for International Organizations of Medical Sciences (CIOMS) and were also in compliance with the laws, regulations, and policies of the "Animal welfare assurance for humane care and use of laboratory animals," permit number A5761-01 approved by the Office of Laboratory Animal Welfare (OLAW) of the National Institutes of Health, USA.

### Brain tissues

Brains of the different animal species used in this study were collected from various institutions and tissue banks, provided with the necessary authorizations (see Ethic statement and S1 Table).

*Mus musculus* (Mouse) and *Oryctolagus cuniculus* (Rabbit) brains came from the NICO animal facility (Orbassano, Turin, Italy). *Heterocephalus glaber* (Naked mole rat; NMR) from the School of Biological and Chemical Sciences, Queen

Mary University of London, London. *Callithrix jacchus* (Marmoset) brains were provided by the University of Zurich, Switzerland and from the Department of Anthropology, The George Washington University, Washington, DC, USA. *Felis catus domestica* (Cat) and *Equus caballus* (Horse) brains came from the Department of Comparative Biomedicine and Food Science, University of Padova, Italy. *Ovis aries* (Sheep) came both from the Department of Comparative Biomedicine and Food Science, University of Padova, Italy, and from the INRA research center, Nouzilly, France. *Pan troglodytes* (Chimpanzee) comes from the National Chimpanzee Brain Resource (chimpanzeebrain.org).

Four prepuberal mouse (*M. musculus*; C57BL/6 mice raised at the NICO facility) brains were extracted a few minutes following euthanasia and fixed by immersion. Four young adult, four middle-aged, and four aged mouse were used. For these brains, transcardiac perfusion was performed under anesthesia (i.p. injection of a mixture of ketamine, 100 mg/kg, Ketavet, Bayern, Leverkusen, Germany; xylazine, 5 mg/kg; Rompun) with 4% paraformaldehyde in 0.1 M sodium phosphate buffer (PB), pH 7.4. Brains were postfixed for 4 h.

Four prepuberal naked mole rat (*H. glaber*) brains were extracted a few minutes following euthanasia and fixed by immersion, while four young adult and four middle-aged animals were intracardially perfused with 4% paraformaldehyde in 0.1 M sodium PB, pH 7.4 (Dr. Chris G. Faulkes, London, UK). Brains were then postfixed overnight.

Three adult marmosets (*C. jacchus*) were obtained from the Veterinary service of the University of Zurich. The brains were extracted postmortem and fixed by immersion in 4% PFA with 15% picric acid (PA) for 24 h. The exact ages of the animals were unknown; they were aged as adults (categorizable in the middle age life stage) by an experienced veterinarian with the following criteria: the closure of the femoral and humeral epiphyseal plate, the body weight, the forearm length and sexual maturity. Four young adults and one middle-aged marmoset were provided by the Texas Biomedical Research Institute (USA): they were collected at the time of necropsy following euthanasia. All brains were immersion-fixed in 10% buffered formalin immediately at necropsy. After a 5-d period of fixation, brains were transferred into a 0.1 M phosphate buffered saline (PBS, pH 7.4) solution containing 0.1% sodium azide and stored at 4° C. None of the brains included in this study showed gross abnormalities or pathology on veterinary inspection.

Four prepuberal and four young adult rabbit (*O. cuniculus*) brains come from a stock at the NICO animal facility used in a previous report [82]. Animals were deeply anesthetized (ketamine 100 mg/kg – Ketavet, Bayern, Leverkusen, Germany – and xylazine 33 mg/ kg body weight - Rompun; Bayer, Milan, Italy) and perfused intracardially with a heparinized saline solution followed by 4% paraformaldehyde in 0.1 M sodium PB, pH 7.4. Brains were then postfixed for 6 h.

The brains of four young adult and four middle-aged cats (*F. catus domestica*), four young adult sheep (*O. aries*), four young adult and four middle-aged horses (*E. caballus)* were provided from the University of Padova (tissue samples preserved in the Mediterranean Marine Mammal Tissue Bank - MMMTB - are distributed to qualified research centers worldwide, following a specific documented request, according to national and international regulations on protected species - CITES). These brains, obtained postmortem, were fixed by immersion in 10% buffered formalin and kept in the fixative solution for one month (sheep, cat), and 3 months (horses). Sheep and horse samples originated from a commercial slaughterhouse and were treated according to the European Community Council directive (86/609/EEC) on animal welfare during the commercial slaughtering process and were constantly monitored under mandatory official veterinary medical care. The definition of each age group was based on the official documentation available, corresponding to each ear mark in sheep, and confirmed by teeth examination in horses. Four prepuberal and four middle-aged sheep (*O. aries* - breed: Ile de France) were raised at the Institut National de la Recherche Agronomique (INRA; Nouzilly, Indre et Loire, France; ethical permissions are reported in Ref. [84]. Four prepuberal sheep were perfused through both carotid arteries with 2 L of 1% sodium nitrite in PBS, followed by 4 L of ice-cold 4% paraformaldehyde solution in 0.1 M PB at pH 7.4. The brains were then dissected out, cut into blocks and postfixed in the same fixative for 48 h. Four middle-aged sheep were collected 20 min after death and kept in 10% formalin for 1 month.

Four young adult and four aged chimpanzee (*P. troglodytes*) brains were provided by the National Chimpanzee Brain Resource (USA); they were collected postmortem from Association of Zoos and Aquariums, or National Primate Research

Centers, maintained in accordance with each institution's animal care guidelines and fixed by immersion in 10% formalin [85]. After 10–14 d the brain was transferred in 0.1 M PBS with 0.1% sodium azide solution and stored at 4°C.

**Tissue processing for histology and immunohistochemistry.** After fixation, the whole hemispheres of marmoset, cat, sheep, chimpanzee, and horse brains were cut into coronal slices (1−2 cm thick). The slices were washed in a PB 0.1 M solution, pH 7.4, for 24−72 h (based on brain size) and then cryoprotected in sucrose solutions of gradually increasing concentration up to 30% in PB 0.1 M. Slices of larger brains (chimpanzee and horse) were further reduced in size to be processed in the cryostat or microtome, by cutting them into two blocks, dorsal and ventral. Then, slices, blocks or the entire hemispheres (mouse, naked mole rat, and rabbit) were frozen by immersion in liquid nitrogen-chilled isopentane at −80°C. Before sectioning, they were kept at −20°C for at least 5 h (time depending on brain size) and then cut into 40 μm thick coronal sections using a cryostat (total numbers of sections/hemisphere are reported in S4 Table). Free-floating sections were then collected and stored in cryoprotectant solution at −20°C until staining.

Sections were used both for histological staining procedures (aimed at defining the overall neuroanatomy and boundaries of the entire hemisphere or amygdala for volume estimations) and for immunocytochemical detection of specific markers (S2 Table). Histological analyses were performed on Toluidine blue stained sections (cresyl violet for chimpanzees). For immunohistochemistry, two different protocols of indirect staining were used: peroxidase or immunofluorescence techniques. For 3,3′-diaminobenzidine (DAB) immunohistochemistry, free-floating sections were rinsed in PBS 0.01 M, pH 7.4. Antigen retrieval was performed using citric acid, pH 6.0, at 90°C for 5–45 min. After further washing in PBS 0.01 M, the sections were immersed in an appropriate blocking solution (1%–3% Bovine Serum Albumin, 2% Normal Horse Serum, 0.2%–2%Triton X-100 in 0.01M PBS) for 90 min at RT. Following, sections were incubated with primary antibodies for 48 h at 4°C (S2 Table). After washing in PBS 0.01 M, sections were incubated for 2 h at RT with biotinylated secondary antibodies (anti-goat, made in horse, 1:250; anti-mouse made in horse, 1:250; anti-rabbit made in horse, 1:250, Vector Laboratories). Then, sections were washed with PBS 0.01 M, and incubated in avidin–biotin–peroxidase complex (Vectastain ABC Elite kit; Vector Laboratories, Burlingame, CA 94,010) for 1 h at RT. The reaction was detected with DAB, as chromogen, in TRIS-HCl 50 mM, containing 0,025% hydrogen peroxide for few minutes and then washed in PBS 0.01 M. Sections were counterstained with Toluidine blue staining, mounted with NeoMount Mountant (Sigma-Aldrich, 1,090,160,500) and coverslipped.

For immunofluorescence staining, free-floating sections were rinsed in PBS 0.01 M. Antigen retrieval was performed using citric acid at 90°C for 5–30 min. After further washes in PBS 0.01 M, sections were immersed in appropriate blocking solution (1%–3% Bovine Serum Albumin, 2% Normal Donkey Serum, 1%–2% Triton X-100 in 0.01M PBS) for 90 min at RT. Then the sections were incubated for 48 h at 4°C with primary antibodies (S2 Table), and subsequently with appropriate solutions of secondary antibodies: Alexa 488-conjugated antimouse (1:400; Jackson ImmunoResearch, West Grove, PA), Alexa 488-conjugated antirabbit (1:400; Jackson ImmunoResearch, West Grove, PA), cyanine 3 (Cy3)-conjugated antigoat (1:400; Jackson ImmunoResearch, West Grove, PA), cyanine 3 (Cy3)-conjugated antiguinea pig (1:400; Jackson ImmunoResearch, West Grove, PA-706-165-148), Alexa 647-conjugated antimouse (1:400; Jackson ImmunoResearch, West Grove, PA), Alexa 647-conjugated antirabbit (1:400; Jackson ImmunoResearch, West Grove, PA 711-605-152) antibodies for 4 h at RT. Immunostained sections were counterstained with 4′,6-diamidino-2-phenylindole (DAPI, 1:1000, KPL, Gaithersburg, Maryland, USA) and mounted with MOWIOL 4–88 (Calbiochem, La Jolla, CA).

**Defining comparable neuroanatomy of the mammalian species.** The mammalian brains analyzed here differ in terms of brain size, gyrencephaly, and overall neuroanatomical organization. In addition to interspecies differences, some other variables were present, mostly due to tissue processing: brains of the same species can differ in single individuals and can undergo variable degrees of shrinkage (e.g., temperature during cutting might affect section thickness). Thus, each specimen in the age groups of each animal species can show slightly different numbers of coronal sections covering the entire hemisphere (and consequently the amygdala). To account for this variable, a fixed number of serial coronal sections covering both the whole hemisphere and amygdala was considered in each species, as follows: the total number of sections obtained for each specimen/species were compared, and the specimen with the lower number of sections was

used as a reference to reach the same number of sections in all specimens. To do this, sections at the very beginning and at the very end of the brain (in the frontal and occipital lobes) were excluded. The same was done for the amygdala, by excluding sections at the very beginning and very end of the structure. In this way, sections were comparable among all specimens of a single species (the number of sections used is reported in S4 Table).

**Volume measurement and cell counting.** **Volume measurement.** To estimate the volume of the brain hemisphere and amygdala region, serial coronal sections at a 480 µm interval apart (1 section out of 12 serial sections; 1 out of 48 in the very large brains of horse and chimpanzee, only for whole hemisphere analysis; see S4 Table for number of sections employed for the analysis) were collected covering the whole hemisphere and amygdala anterior-posterior extension in each species (3 specimen/species at each life stage). The selected sections were stained with Toluidine blue to highlight the overall neuroanatomy (Figs 1C, 1D and S1); the stained sections were scanned using Axio Scan (Zeiss; Oberkochen, Germany) and the whole hemisphere coronal sections and amygdala areas for each specimen were measured using the "Contour Line" tool of ZEN Blue Software (Zeiss; Oberkochen, Germany). The volume was calculated in each specimen by using the following formula:

Mean of area of hemisphere coronal sections or amygdala × Section thickness (0.04 mm) × Number of sections covering the entire hemisphere or amygdala length in a single specimen.

**Cell counting.** To perform the quantitative analysis of both DCX$^+$ cells and Ki67$^+$ nuclei, serial sections at a 480 µm interval from each other were collected, covering the whole amygdala in each species (4 specimens/species at each life stage; Figs 1 and S2). The counting was done using Neurolucida software (MicroBrightfield, Colchester, VT) as follows: a trained operator performed the tracing of the amygdala area in each section using the "Contour" tool of Neurolucida and all positive stained cells were counted with markers using a 20x magnification objective lens (S2C,D Fig) to obtain the DCX$^+$ or Ki67$^+$ cell density/mm$^2$. Cells cut on the superior surface of the section were not considered, to avoid overcounting. The cell soma size was obtained by evaluating the cell soma width (diameter orthogonal to main axis), measured in about 100 cells for each animal species using the Neurolucida 'measure line' tool.

Identification of possible Ki67$^+$/DCX$^+$ double-stained cells was performed using a 40x magnification objective lens with a confocal microscope (Nikon Eclipse 90i microscope - Nikon, Melville, NY). For all specimens (aside from chimpanzees), 3 sections from the anterior, middle, and posterior part of the amygdala were selected and, in each of them, three microscopic fields in the dorsal, medial, and ventral part of amygdala were analyzed.

The percentage of NeuN$^+$/DCX$^+$ double-stained cells was measured in marmoset, rabbit, and cat (3 specimens/species). For each specimen, 3 sections from the anterior, middle, and posterior part of the amygdala were selected and, in each of them, three microscopic fields including both type 1, type 2a, and type 2b cells were acquired. A total of 27 confocal fields/species and at least 300 cells/species were counted for the analysis (tabulated data can be found in S1 Data).

To evaluate the percentage of immature cells in each of the amygdala subnuclei, three representative sections/species employed for DCX$^+$ cell counting at young adult age (belonging to three amygdala levels: anterior, middle, and posterior) were used. Each amygdala subnucleus was drawn from the slide using Neurolucida software, with markers placed on DCX$^+$ cells. The resulting contour and cell markers were converted to an image file and imported to ImageJ and the percentages of area occupied by the immature cell markers (area fraction; the percentage of pixels in the selection that have been highlighted in red; see Fig 6C and S3 Table) was measured in each of the amygdala subnuclei (in its anterior, middle, and posterior part) by using the Image/Adjust/Threshold function of ImageJ.

The analysis of DCX$^+$ cells with migratory morphology was performed on two individuals of all species (young adult age). For each specimen, two DAB-stained sections were used for cell counting (see above), selected from the anterior, middle, and posterior part of the amygdala; the sections were scanned using Zeiss Axio Scan.Z1 (objective: 20x magnification). In each scan, the whole amygdala was analyzed using Zen Blue Software in search of DCX$^+$ cells with migratory morphology (small, oval/fusiform cell soma with an unbranched leading process, as described in Sorrells and colleagues [30] and Alderman and colleagues [7]; S7 Fig). Cells corresponding to this morphology were marked using the "arrow"

marker of Zen Blue Software (the tip of the arrow indicating the direction of the leading process). To obtain the final percentage, the total number of migratory-like DCX$^+$ cells was divided by the total number of DCX$^+$ cells of the analyzed section (number obtained from the previous cell counting analysis, see above) and multiplied by 100.

The number of sections cut in each hemisphere and those considered for counting procedures in each species are listed in S4 Table. Overall, a total of 4,413 cryostat sections coming from 80 brains were analyzed.

**Image acquisition and processing.** Images were collected using a Nikon Eclipse 90i microscope (Nikon, Melville, NY) connected to a color CCD Camera, a Leica TCS SP5, Leica Microsystems, Wetzlar, Germany, and a Nikon Eclipse 90i confocal microscope (Nikon, Melville, NY). For volume analysis and analysis of cell migration, Zeiss Axio Scan.Z1 and Zen Blue software were used (Zeiss; Oberkochen, Germany). For quantitative analysis of DCX$^+$ and Ki67$^+$ cells, Neurolucida software (MBF Bioscience, Colchester, VT) was used. For both Ki67$^+$/DCX$^+$ and NeuN$^+$/DCX$^+$ double-stained cell analyses, "Cell Counter" Plugin of ImageJ software (version 1.50b; Wayne Rasband, Research Services Branch, National Institute of Mental Health, Bethesda, Maryland, USA) was used.

All images were processed using Adobe Photoshop CS4 (Adobe Systems, San Jose, CA) and ImageJ version 1.50b (Wayne Rasband, Research Services Branch, National Institute of Mental Health, Bethesda, Maryland, USA). Only general adjustments to color, contrast, and brightness were made.

The percentages of areas occupied by the immature cells (area fraction; the percentage of pixels in the selection that have been highlighted in red; see Fig 4D) in each of the amygdala subnuclei (and in each of its anterior, middle, and posterior part) were measured by using the Image/Adjust/Threshold function of ImageJ imported from the Neurolucida fields employed for DCX$^+$ cell counting.

**Statistical and phylogenetic analysis.** All graphs and relative statistical analysis were performed using GraphPad Prism Software (San Diego, California, USA). Since our data were not normally distributed, we used different nonparametric tests: Mann-Whitney test (two-tailed), Kruskal-Wallis test with Dunn's multiple comparison post-test. $p < 0.05$ was considered statistically significant. Medians were used as data central measure as previously done for the analysis of INs in the cerebral cortex [15].

Species mean DCX$^+$ and Ki67$^+$ cell densities were used to perform ancestral character state reconstructions of trait evolution mapped onto the phylogeny. A phylogenetic tree of the species in the sample was downloaded from the Time-Tree database [86]. The ancestral character state reconstruction was implemented in Mesquite software (version 3.81), using a parsimony model.

To determine the scaling relationships in our dataset, we employed least squares regression. All data were log transformed to fit power functions to linear regression, as is standard procedure in comparative studies of neuroanatomy.

## Supporting information

**S1 Fig. Subnuclei segmentation in the amygdala of different mammals.** (**A**) Amygdala subnuclei segmentation was performed on coronal sections stained with toluidine blue; cresyl violet in chimpanzees and matched with: ([7,37,38] mouse; [39] naked mole rat; [40,41] rabbit; [28,42] cat; [43,44] sheep; [45–47] marmoset; [47,48] chimpanzee; [49] horse). Due to heterogeneous and fragmentary literature data concerning the subnuclei segmentation in different species, a simplified delineation and grouping were adopted (here shown in mouse and primates; for all species and different anterior-posterior levels, see Fig 2) to be used for studying the number and distribution of DCX$^+$ cells in all eight species considered here. La, Lateral nucleus; Ba, Basal nucleus; Ab, Accessory Basal nucleus; P, Paralaminar nucleus; Ce, Central nucleus; Me, Medial nucleus; Co, Cortical nucleus; I, Intercalated nuclei; St, Stria terminalis. Scale bars: 500 µm. (TIF)

**S2 Fig. DCX$^+$ cell quantification.** (**A**) The entire amygdala's anterior-posterior length of each species (from 1.4 to 9.6 mm) was analyzed immunocytochemically in sections at 480 µm apart (one serial coronal section out of 12; from 3

sections in rodents to 20 sections in horse). (**B**, **C**) After establishing amygdala areas through segmentation carried out on serial coronal sections (see Figs 1 and 2), DCX⁺ cells (red crosses) and Ki67⁺ nuclei (dividing cells; black dots) were counted with Neurolucida software (example given in **D**), to obtain cell density/area and cell density/volume of the entire region of interest. The anterior-posterior distribution and the bidimensional spatial distribution (dorsal-ventral and lateral-medial) of immunoreactive elements in single coronal sections were also studied (see Figs 6 and 7). While counting the DCX⁺ cells, a distinction was made between type 1, type 2a, and type 2b (the latter then considered together as type 2 cells), based on their soma size and dendritic arborization (**D**; see also Fig 3A). The aim was to obtain comparable cell densities and spatial distributions in all species to investigate phylogenetic variation (and age-related variation), and to check for presence/absence/amount/nature of dividing cells (see text and Table 1). Colors for the two main cell types (yellow and brown) are the same in pie charts of Fig 5C).
(TIF)

**S3 Fig. Internal positive controls in the cerebral cortex layer II (piriform cortex; age 41, 6 y; see also the study by La Rosa and colleagues [15], in which the same anti-DCX antibody was employed) and amygdala (basolateral complex; age 31, 6 y here immunostained with a different antibody) of an adult chimpanzee brain.** Note the densely packed network of DCX⁺ cells in the amygdala and the absence of staining in the external capsule (Ec). Scale bars: 50 μm.
(TIF)

**S4 Fig. Higher magnification of** Fig 3A **(Tbr1 staining) showing the separate channels for each marker: DAPI, Doublecortin - DCX -, and Tbr1.**
(TIF)

**S5 Fig. Spatial distribution in the amygdala anterior-posterior (longitudinal) axis.** The amount of DCX⁺ cells (**A**, density/area; red) and Ki67⁺ nuclei (**B**, black) are reported as histograms (y-axis) for each single coronal section of the amygdala counted in each species (from 3 in rodents to 20 in horse, x-axis; see also Fig 6A), in the sequence from the anterior (left) to posterior part (right), thus revealing the pattern of longitudinal spatial distribution of these cell populations. While the distribution of the DCX⁺ cells is highly heterogeneous among mammals (see also Fig 6A), e.g., far higher in the anterior (mouse), middle (primates), or posterior part (sheep, cat), that of dividing cells is quite homogeneous, especially in gyrencephalic species and marmoset. Animal species are arranged from top to bottom according to increasing DCX⁺ cell density (mean cell density in the whole amygdala). Tabulated data can be found in S1 Data. Brain icon from https://app.biorender.com/.
(TIF)

**S6 Fig. Enlargement of plots A and B in** Fig 8 **(DCX⁺ and Ki67⁺cell densities in the amygdala at different ages) using the color code for ages (see** Fig 1**).**
(TIF)

**S7 Fig. Analysis of DCX⁺ cells showing migratory morphology (cells with small, oval/fusiform cell soma – less than 9 μm – and an unbranched leading process) in the amygdala of all species considered, at the young adult stage.** Red arrows indicate the orientation of the leading processes. **A**, Examples of DCX⁺ cells with migratory morphology in the amygdala of different species. At low magnification, only some of these cells are detectable (white arrows) out of many immature cells. **B**, The amount of these cells (percentage with respect to all DCX⁺ cells in the section), their topographical location in the anterior, middle, and posterior part of the amygdala, as well as the direction of their leading-like process, is shown. Green line, amygdala perimeter. The percentage of migratory-like cells (pie charts) decreases going from rodents to horse and primates, their direction being mostly random in all species considered, thus showing no specific alignment. Scale bars: immunofluorescence, 30 μm; diaminobenzidine staining, 20 μm.
(TIF)

**S1 Table. Animals and brain tissues used in this study (4 specimens/age).** (a) Institute of Anatomy – University of Zurich; (b) Department of Comparative Biomedicine and Food Science – University of Padova; (c) National Chimpanzee Brain Resource – USA; (d) Neuroscience Institute Cavalieri Ottolenghi (NICO); (e) INRA research center – Nouzilly, France; (f) School of Biological and Chemical Sciences, Queen Mary University of London, London; (h) George Washington University, Washington DC. PMI: postmortem interval; CA, carotid artery; IC, intra-cardiac; PFA, paraformaldehyde solution. Age groups: PP, Prepuberal; PA, Picric acid; YA, Young adult (in bold since a stage present in all species); MA, middle age; AG, aged; *see text; d, days; m, months; y, years. IC, intra-cardiac; CA, carotid artery. Animal species are arranged from top to bottom according to their increasing brain size. For information regarding ages the Animal Diversity Web ([87]; available at https://animaldiversity.org/) was used.
(DOCX)

**S2 Table. Primary antibodies used in this study.**
(DOCX)

**S3 Table. Mean percentages of areas occupied by DCX$^+$ immature neurons in the amygdala subnuclei.** La, lateral nucleus; Ba, basal nucleus (*the PL has been included both for species in which it has been previously described and in the others, wherein it is considered as part of the Ba); Ab, accessory basal nucleus; PL, paralaminar nucleus; BLc, basolateral complex (including the PL); Ce, central nucleus; Me, medial nucleus; Ce-Me, centro-medial nucleus; Co, cortical nucleus. Background colors (dark gray, white, and light gray) identify the three amygdala subdivisions shown in Fig 7B.
(DOCX)

**S4 Table. Number of cryostat sections cut in each hemisphere and used for quantitative analyses in the different animal species and ages.**
(DOCX)

**S5 Table. Estimation of the total number of DCX$^+$ cells in the amygdala of different mammals (one hemisphere).**
(DOCX)

**S6 Table. Estimation of the total number of Ki67$^+$ cells in the amygdala of different mammals (one hemisphere).**
(DOCX)

**S1 Data. All tabulated data.**
(XLSX)

## Acknowledgments

We thank Irmgard Amrein for her generous gift of adult marmoset brains, Cell Signaling Technology for generous gift of rabbit monoclonal anti-DCX antibody (40,619), Ugo Ala for precious advice in statistical analyses, Alessia Pattaro, Alessandro Zanone, Eleonora Pintauro, Elaine Miller, and Dustin Howard for technical help in some experimental procedures, and Enrica Boda for stimulating discussion on glial cells.

## Author contributions

**Conceptualization:** Luca Bonfanti.

**Data curation:** Marco Ghibaudi.

**Formal analysis:** Marco Ghibaudi, Nikita Telitsyn, Chet C. Sherwood.

**Funding acquisition:** Chet C. Sherwood, Luca Bonfanti.

**Investigation:** Marco Ghibaudi, Chiara La Rosa, Jean-Marie Graïc, Chris G. Faulkes.

**Methodology:** Marco Ghibaudi, Chet C. Sherwood, Luca Bonfanti.

**Resources:** Chiara La Rosa, Jean-Marie Graïc, Chris G. Faulkes, Chet C. Sherwood, Luca Bonfanti.

**Supervision:** Luca Bonfanti.

**Visualization:** Marco Ghibaudi, Chet C. Sherwood, Luca Bonfanti.

**Writing – original draft:** Marco Ghibaudi, Chet C. Sherwood, Luca Bonfanti.

**Writing – review & editing:** Marco Ghibaudi, Jean-Marie Graïc, Chet C. Sherwood, Luca Bonfanti.

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
