## [Editor Report · Decision Letter 0]

27 Jan 2025

Dear Dr Bonfanti, 

Thank you for submitting your manuscript entitled "Phylogenetic variation of immature neurons in mammalian amygdala: high prevalence in primate expanded nuclei projecting to neocortex" for consideration as a Research Article by PLOS Biology. Thank you for your patience while we obtained advice from an expert academic editor on your submission.

Your manuscript has now been evaluated by the PLOS Biology editorial staff as well as by an academic editor with relevant expertise and I am writing to let you know that we would like to send your submission out for external peer review. I should note that while we are, in principle, interested in this study, at this stage we have yet to make a firm call about whether the insights provided represent the level of advance that we aim to publish at PLOS Biology. We will therefore be looking for strong reviewer support and enthusiasm in that regard to move forward with the study.

Once your full submission is complete, your paper will undergo a series of checks in preparation for peer review. After your manuscript has passed the checks it will be sent out for review. To provide the metadata for your submission, please Login to Editorial Manager (https://www.editorialmanager.com/pbiology) within two working days, i.e. by Jan 29 2025 11:59PM.

Kind regards,

Taylor

Taylor Hart, PhD, 

Associate Editor

PLOS Biology

thart@plos.org

---

## [Decision Letter · Decision Letter 1]

7 Mar 2025

Dear Dr Bonfanti,

Thank you for your patience while your manuscript "Phylogenetic variation of immature neurons in mammalian amygdala: high prevalence in primate expanded nuclei projecting to neocortex" was peer-reviewed at PLOS Biology. It has now been evaluated by the PLOS Biology editors, an Academic Editor with relevant expertise, and by several independent reviewers. 

In light of the reviews, which you will find at the end of this email, we would like to invite you to revise the work to thoroughly address the reviewers' reports.

I've pasted the full reports underneath my signature. You’ll see that all three reviewers are quite positive, praising the study's quality and describing the findings as interesting and relevant. However, they have each provided important suggestions to further strengthen the study, and we think their comments should be thoroughly addressed before we can consider your paper for publication. 

Based on this, we would like to invite a Major Revision of the study. After discussion with the Academic Editor, we think that the revised submission should include new analyses to address the possibility of DCX neuron migration as outlined by R3. The revision should also carefully consider the concerns over cross-species positional matching, accounting for differences in growth and shape of the region, as discussed by R2. The revision should also include textual changes to improve overall clarity, and you should address all other points raised by the reviewers.

Given the extent of revision needed, we cannot make a decision about publication until we have seen the revised manuscript and your response to the reviewers' comments. Your revised manuscript is likely to be sent for further evaluation by all or a subset of the reviewers.

**IMPORTANT - SUBMITTING YOUR REVISION**

*Re-submission Checklist*

*Published Peer Review*

*PLOS Data Policy*

*Blot and Gel Data Policy*

Sincerely,

Taylor

Taylor Hart, PhD, 

Associate Editor

PLOS Biology

thart@plos.org

REVIEWS:

Reviewer #1: This study by Ghibaudi et al. builds upon previous work by the research group of Luca Bonfanti, dealing with the characterization of so-called "immature neurons" in different brain regions and different species. "Immature neurons" in this context stand for cells that fulfill the criteria for immature neurons as they occur during neurogenesis, but are found in brain regions in which no de novo neurogenesis from resident or distant precursor cells occurs. The group has developed clear criteria for this population of cells and these criteria are also applied in this new study.

Generally, this research has already added an important and very interesting facet to the discussion of neurogenesis in the adult brain, highlighting that the presumed functional benefits of immature neurons are not dependent on stem cell proliferation and the complete progression through all stages of neurogenesis. The focus of the present descriptive study is on these cells in the amygdala of different species, finding that these cells are more prevalent in primates than in other species. While immature cells, e.g. in the form of doublecortin (DCX) expressing cells, have been described for the amygdala before, and there have been, albeit highly controversial, reports on complete adult neurogenesis in the amygdala, the subject has never been studied thoroughly and especially not in the light of the insight that Bonfanti and colleagues have provided for this intriguing population of neuronal cells.

The submitted manuscript now offers a very detailed and in-depth description of immature neurons across different species. The presentation of the results is excellent and the figures are very appealing. The depth of description is outstanding. It sets and passes neuroanatomical standards rarely seen in publications today. The authors dare to be descriptive, which is also rare, but occasionally necessary. The paper lays a thorough foundation for other studies that might make attempts to address these cells from a functional or molecular perspective.

The key bird's-eye-conclusion from the analysis is that considerable differences exist between different species, both with respect to the distribution of the doublechortin-positive and of proliferating cells in the amygdala. The authors make a very strong case for their conclusion that the described cells in the amygdala are indeed neurons arrested in maturation or immature neurons, extending their past description for the neocortex to basal brain structures. The most interesting finding is that it seems that this particular population of cells with its implications for plasticity are more abundant in primates than in other mammals. 

The data as they are presented raise little questions or concerns and the general conclusion is easy to follow. The only major criticism is that the text is a bit too rich in words and often not very clear. Already the title appears somewhat convoluted. The text could be tightened without too much effort and the text reworked to obtain a clearer narrative and flow. It could be more to the point. The abstract could be more detailed and more informative. The discussion of how the concept of immature neurons fits into ideas of general brain plasticity could be much clearer. 

Reviewer #2: This is a very interesting manuscript on the presence of immature neurons (INs), identified by immunoreactivity for cytoskeletal protein doublecortin (DCX), in the amygdala across eight species of mammals, including species with a small, lysencephalic brain (like the mouse, the naked mole rat and the rabbit) to species with a larger brain and different degrees of gyrencephaly (cat, sheep, horse, chimpanzee). They also include the marmoset, a small primate with a mostly lysencephalic brain. The authors undertook a systematic investigation across these eight species, from prepuberal to aged animals, in order to assess phylogenetic variation in the presence and abundance of INs, and compare the expression of DCX with markers of cell proliferation, such as Ki67, and markers of neuronal immaturity (adhesion molecule PSA-NCAM), neuronal differentiation (NeuN), and oligodendroglia (Sox10, Olig2) to evaluate the nature of the DCX+ cells. They find expression of Ki67 in scattered cells of the amygdala adult animals (more in rodents than in primates), but with a distribution pattern different from that of DCX+ cells, and they never find coexpression of DCX and Ki67, but they see coexpression of Ki67 and oligo makers, concluding that proliferation in adult amygdala is basically related to production of glial cells, but not neurons. Based on coexpression of DCX and PSA-NCAM and less frequently DCX and NeuN, the authors conclude that the DCX mostly include immature neurons at different degrees of maturation. They propose that these DCX cells represent a reservoir of non-dividing immature neurons of the amygdala (as found in the neocortex), which become gradually more abundant and stable with age in animals with large gyrencephalic brains and, specially, in primates (both lysencephalic and gyrencephalic). These INs with 'arrested maturation' appear to represent a source of plasticity in animals with high social cognition abilities and learning demands. This type of studies are highly relevant as they contribute to a better understanding of brain plasticity (including that in the human brain) and here the authors present high quality material supporting their major findings. However, I have the following concerns and provide some comments and suggestions that may help to improve the impact of the study.

Major concerns:

1. Topography versus topology: In the results the authors provide a 'topographical' distribution of the DCX+ cells and compare it across species. However, the use of topographical coordinates is highly inadequate for comparing across species with different degrees of brain growth and gyrencephaly. For example, what is lateral in the amygdala of a mouse, it becomes ventral or even ventromedial in a monkey. Thus, I suggest to change to 'topological coordinates', which are the natural coordinates that relate to the internal axes. This will provide a better understanding of the comparison of the DCX across species. As an example, in page 20 and figure 5 legend, the authors say that DCX+ cells tend to locate in 'ventral' or 'ventromedial' positions. In which species? These are topographic terms and the position clearly change depending on the species brain being analyzed. It would be more useful to use some clear landmark to refer to the position, such as if the cells are adjacent to the amygdalar capsule (sometimes referred to as the external capsule). Another possibility is to start talking with precision about the amygdala subdivisions where these cells are located (see next point), as when looking at the images it appears that there is a trend fo the DCX cells to be found in or around the so-called paralaminar nucleus (as seen in Fig. 4C, in topographical terms this nucleus is lateral in mouse, but ventral in marmoset and chimpanzee). If the authors do not like the term paralaminar (although they represent it in the Fig. 4C), they can stay with the Ba or BLc (mentioning, for example, that the cells are mostly at the external border, next to the amydalar capsule), and at least readers will know better where the cells locate.

2. Amygdala subdivisions: These are represented in Fig. 4, although the authors do not refer to them until too late in the Results. For example, they do not mention these in Figure 5, or the description in the text related to Fig. 5. I think that amygdalar subdivision should be present in the description of the results from the very beginning, and the images shown in Figure 4 are a start point. However, the delineation of the nuclei represented in Fig. 4C require revision. For example, in mouse, Ce is including the intercalated cells, La is including part of the dorsal endopirifom nucleus, Me is including part of the intercalated cells and the intraamygdaloid subnucleus of BST, but its ventral part has erroneously been assigned to Co. Please use the Paxinos atlas to improve delimitation of nuclei. The same is true for the primate amygdala nuclei (specially problematic is the delineation of the medial amygdala). The authors can refer to the study by Barger et al., 2012, in J. Comp. Neurol (JCN 520:3035-3054; one of the authors is co-author in the present manuscript), and for the medial amygdala the paper by Ghashghaei and Barbas, 2002, Neuroscience Vol. 115, No. 4, pp. 1261-1279).

Other comments:

3. Fig.1: What means the pie-chart in A (top left)?

4. Results (page 14 of PDF), second paragraph, line 3: Regarding DCX cells 'being absent in all coronal section', do you mean being scarce, instead of absent? If you say absent, it would be incoherent with saying that there were 'only a few, scattered DCX+ cells', as stated in this same sentence and shown in Fig. 2.

5. Fig. 2: Please explain the fluorescent staining better: in red DCX, in blue DAPI? (the same happens in other figures). Also, what are you pointing with the arrowheads: a specific morphological subtype of DCX+ cell, or just examples of DCX+ cells? In addition, the amygdala is extremely complex, what part of the amygdala are you showing here? See comment above about amygdala subdivisions. Moreover, it would be relevant to mention somewhere that, from a developmental point of view, the amygdala of all amniotes contains a pallial part (including the basal nuclear complex and cortical areas, rich in glutamatergic neurons), a subpallial part (including the central nucleus, rich in GABAergic neurons), and another division (the medial nucleus) that contains a mix of GABAergic and glutamatergic neurons with different embryonic origins in the pallium, subpallium and the telencephalon-opto-hypothalamic domain. It is interesting to know the location of the DCX cells and the Ki67 cells with respect to these major subdivisions.

6. Results (page 16 of PDF), regarding 'their glutamatergic identity was confirmed by co-expression with the excitatory neuron transcription factor T-box brain 1 (Tbr1; 38, 39; Fig. 3A': To really appreciate coexpression, it would be important to show the separate chanels (for each marker), in addition to the merged image.

7. Fig 3: The red arrowheads in A (in left image) are very difficult to see. I suggest to change the color or the tone to improve visibility.

8. Results (page 20 of PDF, line 1), regarding 'In Fig. 5A are reported the results for quantifications in young adult animals…': suggest to move 'are reported' to after 'animals'

9. Fig 7: In addition to the relative area occupied by the DCX+ cells, another value that could be interesting to show is the density of the DCX+ per mm2 in that area. For example, cat and rabbit show similar area percentage occupied by DCX+ cells, but they appear to be very different in density of DCX+ cells in the paralaminar nucleus. At the end of the Figure legend, the authors say that 'Animal species are arranged from top to bottom according to increasing DCX+ cell density ', but from the images this is not clear in the case of rabbit versus cat, or even with respect to sheep. Both cat and sheep show higher densities than rabbit.

10.Reults (page 27 of PDF), regarding 'the amygdala/brain volume ratio was substantially similar in all species': The authors should consider the paper by Barger et al., 2007 (AMERICAN JOURNAL OF PHYSICAL ANTHROPOLOGY 134:392-403), where they say 'The human lateral nucleus was larger than predicted for an ape of human brain size and occupied the majority of the basolateral division, whereas the basal nucleus was the largest of the basolateral nuclei in all ape species.'

11. Table 1 (in page 30): I suggest to use another format in form of Table, instead of a list a features. For example, for each of the listed findings, you can check if it is so in the different species examined (each species in a different column)

Reviewer #3: In the current manuscript Ghibauti et al perform a comparative analysis of a immature neuron population in postnatal amygdala across a number of species and postnatal ages. Focusing on the basolateral complex, the authors identify several unique features of this neuron population including a much-increased density in larger brained gyrencephalic brains (including primates) and a broader topographic distribution across subnuclei of the basolateral complex than previously identified. 

The work represents a unique and significant contribution to our understanding of the structure and organization of the amygdala. The presence of a population of immature excitatory neurons in mature brain is of high interest to the neuroscience community in general. The unique approach here to evaluate this phenomenon across a wide phylogeny is interesting and could lend important insight to the functional importance of these neurons. Their observation that larger brained highly gyrencephalic mammals seem to contain a much larger pool in amygdala could relate to an increased in the functional demands of these regions in more complex brains. Given the amygdala's contribution to neurodevelopmental disorders such as autism spectrum disorder, this work may also have important implications to disease. 

The work is of high quality and the manuscript is in general, well written. However, there are some general and specific concerns that should be addressed before acceptance. Most of the concerns have to do with the manner in which the data are presented. These are outlined below. 

General Comment 

Throughout the manuscript the authors use commas as decimal points and decimal points instead of commas for reporting numbers. Should this be reversed? 

The authors use the term "coronal plan" throughout manuscript. Should be changed to "coronal plane" 

The authors argue that the relative numbers of scINs is stable in adult primates. But other groups using stereological and non-stereological approaches have shown stark decreases in DCX+ cell number in human paralaminar nucleus (Avillo et al., 2018, Sorrells, 2019) across adulthood. Because paralaminar is not well described across multiple species, they regard the basolateral complex in general and argue the distribution of DCX INs is more widespread in primates than just paralaminar nucleus. Perhaps the reduction in IN numbers in paralaminar previously described in human relates to migration of INs into the basolateral complex as these cells are presumably being integrated into circuitry? The manuscript could benefit from evidence (or lack thereof) of INs migrating into amygdala nuclei from their origin. Could the authors not address this with a more detailed topographic analysis? 

It's somewhat surprising that the percentage of type 2 INs is so small in chimpanzee although they had by far the greatest densities of INs. Given that the progressive insertion of these INs into existing circuits is hypothesized as a strategy for plasticity in higher mammals you might expect to see many more of the more morphologically mature INs in chimpanzee or marmoset. It appears, however, that horse had the most. Can the authors comment on this. Is it reflective of the cognitive demand for amygdala in horse versus primate? As mentioned above, it would be interesting to know the relative topographic location of Type 2 INs versus type 1 if they are in the process of maturing and integrating into mature circuitry. 

Results 

"Characterization of the amygdala DCX+ neuronal population across mammals" 

Paragraph 1, line 1-remove inappropriate "the" 

"Quantification of DCX+ neurons and dividing cells (cell densities) in the amygdala of different 

Mammals" 

3rd paragraph, line 9 "plan" to "plane" 

"Topographical distribution of DCX+ neurons and Ki67+ dividing cells in the amygdala of mammals" 

Last paragraph, changed "As showed..." to "As shown..." 

Materials & Methods 

All known ages should be reported for animals used. 

Figures 

Figure 1 

Pie chart-what does the relative size of each pie refer too. 2 pie pieces aren't labeled, what do these refer to. What do the red shapes refer to in the bottom part of the figure? 

I think the statement "...followed by staining of sections and segmentation of the subcortical region based on histology (B, bottom):" should read "...followed by staining of sections and segmentation of the subcortical region based on histology (C, bottom):" 

Figure 2 

Worry that the labeling in Chimpanzee Amygdala includes off target. How is specificity evaluated in this species? Would be useful to see positive and negative control regions (eg SVZ labeling is in neurogenic niche or also has extra labeling) for chimpanzee. Authors prior evaluation of antibody labeling (Ghibaudi et al, 2023) was on new world monkeys, not chimpanzee. 

Figure 3 

In legend, did you mean to use "douplets" or "doublets" 

Figure 5 

Scale on Y axis of Figure 5b should be modified so the values and difference between species can be better appreciated 

Scales for phylogenetic plots in Figure 5E a, b, c should have values and should be larger. Hard to see even when zoomed in 

Figure 6 

6A: The observations here are interesting. Seems in order to better visualize the comparison across species a normalized line plot would useful so all the species could be plotted on same graph and directly compared. 

6B: similar thing could be useful here. Also, the Y axis values should be adjusted so that the graph values can be seen. For most of the plots it's difficult/imposible to view the values and therefore spatial distributions cannot be seen. 

Axes for plots should have values (eg slice umber, number of cells) 

Figure 7 

Appropriate scale bars should be provided on representative images from each species. 

Figure 8 

For the graphs in A, B, C & D; It is very difficult to distinguish the age groups for given species using the current plot style. Data points associated each age goup should be gven different colors within each graph. For example, in A, for mouse the prepubertal are triangles but it's still difficult to separate them from the other three age groups. If each age group were different colors, it would be easier to distinguish. This should be applied to each of these graphs. 

The level of significance reported in each graph should be reported in the legend. Do 2 symbols (eg A, B; && vs &, D; ** vs *) indicate higher significance? 

Table 1 

Table should be laid out correctly with, title, brief legend, defined rows and columns with headers 

Table S2 

RRID values should be provided for antibodies 

Figure S1 

Sheep posterior outline not numbered

---

## [Decision Letter · Decision Letter 2]

14 Jul 2025

Dear Dr Bonfanti,

Thank you for your patience while we considered your revised manuscript "Phylogenetic variation of immature neurons in mammalian amygdala: high prevalence in primate expanded nuclei projecting to neocortex" for publication as a Research Article at PLOS Biology. This revised version of your manuscript has been evaluated by the PLOS Biology editors, the Academic Editor, and one of the original reviewers.

Based on the reviews and on our Academic Editor's assessment of your revision, we are likely to accept this manuscript for publication, provided you satisfactorily address the following data and other policy-related requests.

IMPORTANT: Please ensure that you address all of the following points:

**Title:

We suggest the following alternate title: Multispecies characterization of immature neurons in the mammalian amygdala reveals their expansion in primates

**Financial disclosure statement:

Please add links to funding agencies in the Financial Disclosure statement in the manuscript details.

**Ethics: 

-- Please ensure that all approval numbers for animal work are included in the Ethic statement.

**Data:

IMPORTANT: Figure 6 appears not to have been uploaded separately (although it is visible in your red-lined manuscript). Please upload the most up to date version of this figure.

--Thank you for providing source data for several of the figures. Please ensure that data are provided for the following figures:

5AB

7B

8ABCDEefgh

S5AB

-- Please cite the location of the data clearly in all relevant main and supplementary Figure legends, e.g. “The data underlying this Figure can be found in S1 Data” or “The data underlying this Figure can be found in https://doi.org/10.5281/zenodo.XXXXX”

-- Please ensure that you are using best practice for statistical reporting and data presentation. These are our guidelines https://journals.plos.org/plosbiology/s/best-practices-in-research-reporting#loc-statistical-reporting and a useful resource on data presentation https://journals.plos.org/plosbiology/article?id=10.1371/journal.pbio.1002128

-- If you are reporting experiments where n ≤ 5, please plot each individual data point.

-- Supplementary files (e.g., excel). Please ensure that all data files are uploaded as 'Supporting Information' and are invariably referred to (in the manuscript, figure legends, and the Description field when uploading your files) using the following format verbatim: S1 Data, S2 Data, etc. Multiple panels of a single or even several figures can be included as multiple sheets in one excel file that is saved using exactly the following convention: S1_Data.xlsx (using an underscore).

-- Please ensure that your Data Statement in the submission system accurately describes where your data can be found and is in final format, as it will be published as written there.

**Code availability:

Per journal policy, if you have generated any custom code during the course of this investigation, please make it available without restrictions. Please ensure that the code is sufficiently well documented and reusable, and that your Data Statement in the Editorial Manager submission system accurately describes where your code can be found. [IF APPLICABLE: As the code that you have generated to XXX is important to support the conclusions of your manuscript, its deposition is required for acceptance.]

**Abstract:

Please note that per journal policy, the model system/species studied should be clearly stated in the abstract of your manuscript. 

--------

We expect to receive your revised manuscript within two weeks. 

*Published Peer Review History*

*Press*

Sincerely,

Taylor

Taylor Hart, PhD, 

Associate Editor

thart@plos.org

PLOS Biology

Reviewer remarks:

Reviewer #3: Authors addressed adequately all of my concerns

---

## [Editor Report · Decision Letter 3]

18 Jul 2025

Dear Dr Bonfanti,

Thank you for the submission of your revised Research Article "Multispecies characterization of immature neurons in the mammalian amygdala reveals their expansion in primates" for publication in PLOS Biology. On behalf of my colleagues and the Academic Editor, Julie Fudge, I am pleased to say that we can in principle accept your manuscript for publication, provided you address any remaining formatting and reporting issues. These will be detailed in an email you should receive within 2-3 business days from our colleagues in the journal operations team; no action is required from you until then. Please note that we will not be able to formally accept your manuscript and schedule it for publication until you have completed any requested changes.

PRESS

Sincerely, 

Taylor Hart, PhD, 

Associate Editor

PLOS Biology

thart@plos.org